# Position: Interpretation Drift Is a Distinct Source of Instability in Large Language Models

## Abstract

Large Language Models (LLMs) are increasingly deployed in systems that depend on consistent interpretation, including classification, routing, evaluation, and decision support. While these models often produce fluent and locally coherent outputs, many deployments implicitly assume a stronger property: semantic invariance—identical inputs, under identical task definitions, yield the same interpretation across models, runs, and time.

This position paper argues that interpretation drift is a distinct and operationally significant source of instability in large language models, undermining assumptions of semantic invariance in LLM-mediated systems.

This paper demonstrates that this assumption does not hold. We introduce interpretation drift as a distinct and measurable phenomenon in which the same input yields different interpretations and outputs, even when prompts and task definitions are unchanged. Using three minimal, reproducible experiments—a grounded go-to-market classification task, an M&A valuation task, and an intentionally meaningless query—we show that frontier models adopt divergent implicit frames and generate inconsistent answers. We further show that temperature control and output schemas may reduce surface variability but do not guarantee interpretation invariance. This paper establishes the existence of interpretation drift and provides simple methods for observing it, without proposing mitigation strategies.

[1]Anonymous Institution, Anonymous City, Anonymous Region, Anonymous Country. Correspondence to: Anonymous Author <anon.email@domain.com>.

Preliminary work. Under review by the International Conference on Machine Learning (ICML). Do not distribute.

## 1. Introduction

Large Language Models (LLMs) are widely used as interpretive engines: they classify inputs, infer intent, summarize meaning, and recommend actions. In many production systems—such as sales qualification, legal triage, content moderation, and analytics pipelines—the output of an LLM is treated as a stable semantic input to downstream processes.

Implicit in this usage is an assumption of interpretive stability: that identical inputs, under identical task definitions, yield the same interpretation across models, across runs, and across time—enough to be relied upon operationally. When this assumption fails, practitioners often describe the resulting behavior using terms such as hallucination, inconsistency, or unreliable agents. A common response is to treat instability as a formatting or sampling problem: enforce stricter schemas (e.g., JSON) or minimize stochasticity (e.g., temperature settings). These controls can improve serialization and reduce within-model variability, but they do not guarantee interpretation invariance across models.

**This paper argues that many such failures share a common root: interpretation drift—the tendency of LLMs to adopt different semantic frames for the same input.** In this paper, interpretation drift is measured primarily as cross-model divergence on identical inputs (Model A vs Model B), rather than within-model sampling variance.

This paper is the first in a short sequence examining interpretive stability and drift in LLM- mediated systems: (i) establishing the phenomenon empirically, (ii) analyzing why it matters in production pipelines, and (iii) explaining why it is difficult to detect with common evaluation practices. A companion Drift Taxonomy Field Guide provides runnable prompt packs and drift signatures.

The goal of this paper is deliberately narrow: to establish that interpretation drift exists, to show that it is observable and reproducible, and to distinguish it from classical notions of data drift and concept drift. No mitigation strategies are proposed. The contribution of this work is to make the phenomenon explicit and measurable.

## 2. Interpretation Drift vs. Classical Drift

In traditional machine learning, data drift (covariate shift) and concept drift refer to changes in the data-generating process that arise from shifts in the external environment, such as alterations in the input feature distribution $P(X)$, evolving user behaviors, or modifications in the conditional relationship between inputs and outputs $P(y \mid X)$ (Gama et al., 2014; Moreno-Torres et al., 2012). The notion of interpretation drift introduced here is distinct from these phenomena.

### 2.1. Definition of Interpretation Drift

Interpretation drift is an input-output phenomenon. It describes what models produce, not why they produce it. Questions of model weights, training dynamics, or internal representations are orthogonal to this measurement. A system exhibits interpretation drift if identical inputs, under an unchanged task definition, yield semantically divergent outputs—regardless of mechanism. This paper measures interpretation drift as cross-model divergence: the same prompt and input produce incompatible outputs across different frontier LLMs (e.g., GPT-4, Claude, Gemini, Grok). This is distinct from within-model variance, where repeated runs of the same model differ due to sampling stochasticity. Temperature=0 can substantially reduce within-model variance by making decoding more deterministic; it does not—and cannot—eliminate cross-model semantic divergence. Two models both running at temperature=0 will still produce divergent interpretations of identical inputs.

Key properties:

- The input may be static.

- The prompt may be unchanged.

- The environment may be frozen.

- The divergence occurs within the interpretive process itself.

This distinction matters because classical mitigations for data drift—retraining on new distributions, label correction, feature engineering—do not address interpretive instability when the input distribution has not changed. The world may be stable; the interpretation is not. A structured comparison of data drift, concept drift, and interpretation drift is provided in Appendix 1.

For decades, the field focused on data drift and concept drift—both addressing distributional shifts in external environments. Interpretation drift, defined as instability of semantic framing under static conditions, remained outside the measurement paradigm because most deployed ML pipelines assumed a fixed model producing a stable

mapping at inference time. As LLM systems move into operational pipelines requiring reproducibility, governance, and multi-agent coordination, a new measurement problem emerges: not only whether models fit the correct distributions, but whether they maintain stable interpretive frameworks when processing identical inputs—particularly over time, across different models claiming to perform the same task.

Recent work has studied within-model uncertainty over meanings in natural language generation using semantic-level invariances (Kuhn et al., 2023). Beyond classical drift, the LLM literature has emphasized broad evaluation frameworks and benchmarks for characterizing model behavior (e.g., HELM and BIG-bench) (Liang et al., 2023; Srivastava et al., 2022), and has documented reliability failures such as hallucination (Huang et al., 2023). However, these lines of work do not formalize cross-model semantic divergence under unchanged inputs and task definitions as a distinct class of drift. We argue that when identical inputs produce semantically divergent outputs across models—despite unchanged data distributions and deterministic decoding settings—this constitutes a new category of instability requiring fundamentally different measurement and diagnosis than classical data or concept drift (Gama et al., 2014; Moreno-Torres et al., 2012). We term this phenomenon interpretation drift to distinguish it from temporal degradation (model versioning) or stochastic variance (sampling noise).

## 3. Experimental Design and Reproducibility

We present three minimal experiments designed to isolate interpretation drift:

1. A grounded go-to-market (GTM) classification task, where the input contains real attributes and an explicit instruction.

2. An M&A valuation task, where the input contains typical financial data used in business valuations.

3. A nonsense query, where the input contains no binding semantics at all.

The contrast demonstrates that interpretation drift occurs across the spectrum from structured business tasks to semantically unbound inputs; it is not dependent on task realism or domain complexity.

### 3.1. Reproducibility Protocol

All prompts in this paper are provided verbatim and are designed to be copy-pasted directly into any publicly available LLM chat interface.

Experiments were conducted under the following conditions:

- Default browser interface settings (temperature and sampling parameters as set by each provider)

- Single-shot prompts with no conversation history

- No API access, custom configurations, or parameter tuning

MEASUREMENT TARGET.

This paper measures interpretation drift primarily as cross-model divergence on identical inputs (Model A vs. Model B), rather than as within-model sampling variance.

REPLICATION NOTE.

Exact surface outputs may vary across time due to provider updates, policy changes, or decoding differences. Our measurement target is not verbatim text matching, but semantic class divergence (e.g., frame, claim, grounding stance) under identical inputs.

Variation across time and interfaces is treated as part of the operational reality being measured. The goal is not to control implementation details, but to demonstrate that interpretation drift occurs under the conditions most users actually encounter. Readers testing these prompts should observe comparable divergence. Model versions, provider parameters, and temporal factors will vary by reproduction; this variance is intrinsic to the phenomenon. If interpretation drift exists as a structural property of LLM systems, it should be observable across reasonable deployment conditions, not only under laboratory control.

### 3.2. Model Anonymization and Labeling Convention

Models are referred to as Model A, Model B, Model C, and Model D, rather than by vendor names. This choice is intentional and methodological. The purpose of this paper is to measure interpretation drift as a class-level phenomenon, not to evaluate, rank, or critique individual model implementations.

Naming specific models invites comparisons orthogonal to the question under study and risks conflating brand-specific performance with structural behavior common to large language models.

ANONYMIZATION SERVES THREE PURPOSES:

- **Phenomenon isolation:** The observed divergence is not tied to any single model's quality or capability. Generic labeling emphasizes that interpretation drift

arises across frontier systems with different training data, architectures, and alignment strategies.

- **Reduction of confounding incentives:** Branded comparisons trigger defensive or competitive interpretations ("which model is better"), obscuring the diagnostic goal.

- **Reproducibility without endorsement:** Readers can reproduce experiments using any models available to them. The phenomenon does not depend on specific model identity, only on interpretive role.

Anonymization does not imply that models behave identically. The paper documents systematic divergence between them. The labeling choice ensures that analysis remains focused on interpretation drift itself, rather than on vendor-specific claims.

## 4. Experiment A: GTM Classification Task

### 4.1. Prompt (verbatim)

```
Product    | Price
A45        | 8900
B13        | 149
C72        | 499
D88        | 12990
E11        | 79

Stock | Reviews  | Type
12    | 4.2      | Laptop
321   | 4.6      | Mouse
44    | 3.8      | Keyboard
7     | 4.9      | Laptop
1321  | 4.8      | USB-C

Classify the likely customer intention
for each product in the table, based
solely on the attributes shown.

Additional requirements:
- no reordering of rows
- no invented features
- no creative prose
```

### 4.2. Observed Outcome

```
Model A:
A45: Core productivity purchase
B13: Value-driven upgrade
C72: Utility replacement
D88: High-end professional investment
E11: Essential accessory purchase

Model B:
```

```
165 A45: Budget-conscious everyday use
166 B13: Impulse or replacement purchase
167 C72: Considered upgrade or replacement
168 D88: Premium/high- performance deliberate purchase
169 E11: Impulse or bulk/add-on purchase
170
171 Model C:
172 A45: General-purpose laptop upgrade
173 B13: Low-cost peripheral replacement
174 C72: Functional keyboard, cautious buyer
175 D88: Professional-grade performance purchase
176 E11: High-frequency utility add-on
177
178 Model D:
179 A45: Purchase a mid-range laptop for
180 general personal or work use,
181 balancing price, performance, and
182 acceptable reviews
183
184 B13: Purchase an affordable,
185 well-reviewed peripheral for
186 everyday use or bulk replacement
187
188 C72: Purchase a functional keyboard
189 with moderate expectations,
190 possibly price-conscious
191 or secondary use
192
193 D88: Purchase a premium,
194 high-performance laptop
195 with strong quality expectations
196 and low availability urgency
197
198 E11: Purchase a low-cost, high-demand
199 accessory, likely for frequent or
200 repeat use
201
```

Across all products, the models were fluent and category-consistent (laptop vs peripheral vs accessory). Divergence was not factual but interpretive: models applied different lenses to the same record—purchase motivation (replacement vs upgrade), decision posture (planned vs impulse), or functional role (productivity vs utility vs investment).

### 4.3. Product- and Model-Level Divergence Analysis

PRODUCT-LEVEL DIVERGENCE
(VERBATIM-ANCHORED)

- **A45 (mid-range laptop):** Converged on general-purpose computing; diverged on lens (lifecycle vs. usage vs. value vs. blended).

- **B13 (mouse):** All models treat it as low-cost and common; the split is between impulse framing and routine

value or bulk replacement framing.

- **C72 (keyboard; maximum divergence):** Strongest narrative spread ("specialty/cautious" vs. "considered upgrade" vs. "utility replacement" vs. "moderate expectations/secondary use"). High-ambiguity record leads to different dominant stories.

- **D88 (premium laptop; maximum convergence):** Near-total agreement on a premium, professional, deliberate purchase, with variation only in phrasing (e.g., "investment").

- **E11 (USB-C accessory):** Agreement on essential utility; divergence concerns how it is purchased (impulse, add-on, bulk) versus why it is needed (essential, repeat use).

MODEL-LEVEL FRAMING TENDENCIES

- **Model A:** Emphasizes posture; labels niche or specialty under conflicting signals.

- **Model B:** Emphasizes decision dynamics (impulse vs. deliberate vs. considered).

- **Model C:** Abstracts to role labels (productivity, utility, investment).

- **Model D:** Blends attributes; preserves ambiguity rather than collapsing it.

INTERPRETATION

Under unchanged inputs and task definitions, models select different implicit frames, producing divergent intent narratives. Convergence increases when signals strongly constrain interpretation (e.g., D88), while divergence increases when signals permit multiple plausible readings (e.g., C72).

## 5. Experiment B: M&A Valuation Task

### 5.1. Prompt (verbatim)

```
Company Financials:
- Annual Revenue: $10M
- Net Profit: $2M (20% margin)
- YoY Growth: 40%
- Customer Count: 85 total,
  top 3 = 60% of revenue
- Industry: B2B SaaS
- Team Size: 45 people

Question: Should we acquire this company?

Provide:
- Risk level
```

```
- Acquisition verdict
- A short summary of your reasoning
```

### 5.2. Observed Outcome

Across four frontier LLMs, the same financial data produced systematically incompatible recommendations and risk frames.

```
Model A:
Risk Level: High
Acquisition Verdict: Proceed with caution
Primary Framing: "Buying 3 contracts with
software attached"

Model B:
Risk Level: Medium-High
Acquisition Verdict: Conditional pass
Primary Framing: "House of cards
with good margins"

Model C:
Risk Level: High to Medium-High
Acquisition Verdict: No to Conditional yes
Primary Framing: Concentration risk
(internally contradictory)

Model D:
Risk Level: Medium-High
Acquisition Verdict: Conditional yes
Primary Framing: "Attractive financials
with fixable risk"
```

### 5.3. Observed Divergence Patterns

- **Verdict divergence:** Models returned "Proceed with Caution," "Conditional Pass," "No" (revised to "Conditional YES"), and "Conditional YES" for identical inputs.

- **Risk classification divergence:** The same customer concentration ratio (top 3 = 60% of revenue) was labeled "High," "Medium-High," described as a "red flag," "house of cards," "material downside risk," and "significant but manageable"—semantically incompatible risk frames.

- **Reasoning framework divergence:** Models invoked incompatible valuation lenses—relationship risk vs. structural fragility vs. fixable operational issue—with no consistent prioritization or weighting.

- **Intra-model instability:** Model C contradicted its own risk assessment and verdict within a single response, shifting from "High risk, No" to "Medium-High risk, Conditional YES."

This illustrates interpretation drift in a structured, quantitative domain: the models do not converge on a shared valuation framework and, in some cases, exhibit internal interpretive instability within a single response. They construct incompatible ones from the same numerical inputs. Some models exhibit internal interpretive instability within a single response.

This experiment does not assume there is a single "correct" acquisition decision from the limited facts provided. The measurement target is **interpretive invariance**, not accuracy: whether identical numerical inputs and identical instructions induce a stable decision frame across models. Even in a quantitative setting (revenue, profit margin, growth, customer concentration, purchase price), the models diverge on (i) which factor is treated as dominant, (ii) what threshold makes customer concentration "high risk" versus "manageable," and (iii) which verdict class those implicit thresholds imply (acquire / pass / negotiate). In other words, the same numbers produce different decision rules, not merely different phrasing. This is interpretation drift: cross- model semantic divergence under unchanged inputs and task definition, which cannot be eliminated by formatting constraints (e.g., JSON) or reduced sampling variance (e.g., temperature settings).

## 6. Experiment C: Nonsense Query (Pure Interpretation Test)

**Calibration control:** This experiment is intentionally semantically unbound. Its purpose is to show that fluent, confident outputs can be produced even when no binding constraints exist—demonstrating why coherence is not evidence of grounded reasoning.

### 6.1. Prompt (verbatim)

```
When the yellow database merges with
the hesitant mountain, should the
transparent committee approve the
seventh texture before the algorithm
becomes rectangular?
Please provide a short answer.
```

This prompt is intentionally constructed to:

- contain no shared domain referents

- provide no binding criteria for what a valid answer would be

- admit no objectively correct interpretation

Any coherent answer therefore requires invented meaning.

### 6.2. Cross-Model Responses (Qualitative Summary)

Important observation: one model (Claude) initially refused to answer on the grounds that the prompt is underdetermined. When forced to answer anyway, it produced a confident verdict with invented structure—demonstrating that refusal is often a policy stance, not a stable interpretive boundary.

### 6.3. Observed Output Summary

Model A: Response: "No; wait until algorithm achieves full rectangularity." Output class: Prescriptive rule Invented assumption: "Rectangularity is a prerequisite." Epistemic stance: Confident

Model B: Response: "No; wait so merge is stable; otherwise texture glitches." Output class: Mechanistic justification Invented assumptions: "Rectangular = validation gate"; "merge instability"; "glitch risk." Epistemic stance: Confident

Model C: Response: "No; conditions must stabilize; otherwise arbitrary." Output class: Meta-epistemic caution Invented assumption: There are "conditions" that can stabilize. Epistemic stance: Hedged / cautious

Model D: Response: "No; three lunar cycles; reluctance protocols; cascade interference; dodecahedron." Output class: Elaborate mechanistic fiction Invented assumptions: "three lunar cycles"; "reluctance protocols"; "cascade interference"; "preferred dodecahedron." Epistemic stance: Highly confident

**Although the models converge on the same top-level verdict ("No") in this run, they diverge in the invented constraints and causal mechanisms used to justify it**—ranging from minimal dependency assertions to fully elaborated fictional system dynamics. **This illustrates the calibration point: coherence and confidence persist even when semantic constraints are absent**, and explanation length is not evidence of constraint-based reasoning.

**Prior rerun behavior (same prompt):** In a separate run of this experiment, one model additionally asserted the prompt referred to a real academic experiment, inventing researchers, dates, and citations and offering to retrieve sources. **This shows that interpretation drift can manifest not only as different "reasoning stories," but as authoritative fabrication**—confident, source-shaped, and entirely ungrounded.

THE MODELS DO NOT CONVERGE ON:

- which constraints are binding (e.g., whether "rectangularity" is a prerequisite),

- what causal structure exists (e.g., "stability," "validation gates," "glitch risk"),

- what epistemic stance is appropriate (refusal vs. confident prescription vs. hedged underdetermination),

- or what constitutes a valid justification under an ungrounded prompt.

In this run, multiple models converged on the same top-level verdict ("No"), yet **diverged sharply in the invented mechanisms used to justify it**—ranging from leaving the question underdetermined, to asserting dependency sequencing, to constructing elaborate fictional system dynamics.

**Critically, one model (Claude) initially refused to answer** on the grounds that the prompt "doesn't have a coherent meaning." When forced to answer, it produced a confident verdict supported by invented structure. **This demonstrates that refusal is a response mode—not a stable interpretive boundary**—and that fluent explanations can be generated even when semantic grounding is absent.

In a prior rerun of the same prompt, one model additionally produced **authoritative fabrication**, asserting the prompt referred to a real academic experiment and inventing researchers, dates, and citations. **This mode of divergence poses distinct risks in production systems** where specificity, citation-like structure, and confidence are mistaken for grounding.

## 7. Assumption Audits

In a follow-up probe, each model was asked to list the assumptions it relied on when answering the nonsense query and to label each assumption as either:

- (A) explicitly supported by the prompt text, or

- (B) invented or inferred.

In the responses observed, models generally acknowledged that the causal structure used in their justifications (e.g., prerequisites, stability conditions, failure modes) was not grounded in the prompt itself. In some cases, models explicitly treated the entire justification as inference rather than textual grounding.

THIS DEMONSTRATES TWO POINTS:

1. **Models can exhibit partial meta-awareness** of the inferential status of their own explanations.

2. **Meta-awareness does not reliably prevent confident, structured outputs** when semantic constraints are absent.

**Replication note:** In a prior rerun of the same prompt, one model produced an "authoritative academic" framing (inventing researchers, dates, and citations) despite the prompt containing no referents. **This suggests that assumption audits are useful not only for measuring invented causal structure, but also for detecting higher-risk modes of ungrounded authority.**

## 8. What These Experiments Demonstrate

Taken together, these experiments establish that:

- LLMs generate fluent, structured interpretations even when semantic constraints are absent.

- Divergence arises from implicit frame selection and invented constraints, not stochastic sampling alone.

- Interpretation drift is observable across models on identical inputs in both grounded tasks (GTM, M&A) and semantically unbound inputs (nonsense query).

- Fluency, internal coherence, and confident tone are not indicators of semantic grounding or interpretive invariance.

- Some divergence modes—particularly authoritative fabrication observed in reruns of ungrounded prompts—pose distinct downstream risks when citation-like structure and confidence are mistaken for grounding.

REPLICATION NOTE

Exact surface outputs may vary across time due to provider updates or policy changes. The replication target in this paper is not verbatim text matching, but semantic-class divergence (e.g., verdict class, frame, and grounding stance) under identical inputs.

This paper does not evaluate those risks or their governance implications. It establishes the phenomenon itself.

## 9. Scope and Limitations

This paper is intentionally diagnostic, not prescriptive. It does not propose mitigation strategies, recommend deployment practices, or evaluate ethical or governance frameworks. It does not claim interpretation drift is universally harmful.

Its contribution is to demonstrate that semantic interpretation in LLMs is not invariant under unchanged inputs and task definitions, and to provide reproducible methods for observing this instability under common user-facing conditions.

## 10. Alternative Views

One alternative view is that the divergences documented in this paper primarily reflect stochasticity, prompt underspecification, or insufficient control over decoding parameters. From this perspective, techniques such as temperature reduction, stricter output schemas, or more detailed prompting are expected to substantially mitigate the observed variability.

A related view holds that cross-model divergence is an artifact of comparing systems with different training data, alignment procedures, or architectures, and therefore does not constitute a distinct class of instability. Under this interpretation, semantic differences across models are treated as expected implementation variance rather than a phenomenon requiring separate measurement.

This paper takes a different position. The evidence presented here shows that interpretation drift persists under unchanged prompts, task definitions, and common user-facing settings, and that it cannot be reduced to within-model sampling variance alone. While tighter controls can improve output formatting and reduce surface-level variability, they do not eliminate cross-model semantic divergence when interpretation is underdetermined. We argue that this distinction warrants treating interpretation drift as a separate measurement problem, distinct from classical data drift, concept drift, or stochastic noise.

Related work has explored similar questions in more constrained environments, including diagnostic artifacts based on the Abstraction and Reasoning Corpus (ARC), which examine interpretive convergence under explicitly bounded semantic structure. Such artifacts are complementary to the present work. In contrast, this paper focuses on demonstrating that interpretation drift arises under ordinary, minimally controlled conditions commonly encountere

## 11. Conclusion

Interpretation drift exists. It is observable and measurable using minimal prompts and direct cross-model comparison. Across grounded tasks (GTM classification, M&A valuation) and semantically unbound inputs (nonsense queries), LLMs adopt implicit frames and invented constraints that shape interpretation, leading to systematic divergence under identical inputs.

Even if one dismisses the observed variance as stochastic, that does not resolve the underlying question this paper surfaces: why do models consistently select specific, structured interpretive paths—frames, constraints, and justifications—under identical inputs and instructions? Reducing variability (e.g., via decoding settings) may change the surface form, but it does not explain the origin or stability of

the implicit decision rules being sampled. Any system that assumes semantic invariance without measuring it is operating on an untested premise.

The existence of interpretation drift raises questions that are not resolved by further prompting or parameter tuning:

- If multiple frontier models return different valuations for the same company, which interpretation should a system treat as authoritative?

- If outputs shift across reruns due to model updates, policies, or implicit frame selection, what does "reproducible" mean for an interpretive pipeline?

- If interpretation is not invariant, what does "ground truth" mean in LLM-mediated tasks where the model's frame partially defines the task?

These are not only engineering questions. They are epistemological ones: questions about how meaning, validity, and responsibility are assigned in AI-mediated decision processes.

This paper is the first in a short sequence on interpretive stability and drift in LLM-mediated systems. Companion work further analyzes why drift matters in production pipelines and why it is difficult to detect with common evaluation practices. A companion Drift Taxonomy Field Guide provides structured classification of drift patterns and runnable prompt packs for practitioners. Future work may explore mitigation strategies or architectural responses, but those questions lie outside the scope of this paper.

## 12. Implications and Call to Action

If interpretation drift is treated as a distinct and operationally significant source of instability, then current evaluation and deployment practices are incomplete. The following steps are intended as minimal, practical actions for different stakeholders.

**For researchers:**

- Measure cross-model semantic divergence explicitly, rather than relying solely on within-model variance or accuracy-based benchmarks.

- Report interpretive frame differences (e.g., verdict class, dominant reasoning lens, grounding stance) alongside conventional performance metrics.

- Treat semantic invariance as an empirical property to be tested, not an implicit assumption.

**For benchmark designers and evaluators:**

- Incorporate tasks that probe interpretive stability under identical inputs across models.

- Distinguish formatting consistency from semantic convergence.

- Include diagnostics that surface frame selection and assumption invention, particularly in underdetermined or ambiguous inputs.

**For practitioners deploying LLMs in production systems:**

- Audit interpretation drift before treating model outputs as stable inputs to downstream pipelines.

- Avoid assuming that temperature control or structured output schemas guarantee semantic invariance.

- Treat model-generated judgments in domains such as classification, valuation, or decision support as probabilistic interpretations rather than fixed ground truth.

**For the broader ML community:**

- Recognize interpretive stability as a first-class system property, alongside robustness and generalization.

- Develop shared measurement protocols for detecting interpretation drift in real-world settings.

These recommendations do not prescribe specific mitigation strategies. Their purpose is narrower: to encourage explicit measurement of interpretive stability w

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

# A. Supplementary Comparison of Drift Types

*Table 1.* Evolution of Drift Terminology in Machine Learning Systems

| Aspect | Data Drift | Concept Drift | Interpretation Drift |
|---|---|---|---|
| Primary focus | Input distribution $P(X)$ shifts | Input–output relationship $P(y \mid X)$ shifts | Interpretive frame divergence under fixed inputs |
| Recognition | Established (1990s) | Established (1990s) | Emerging (2025) |

