# Empirical Evidence Of Interpretation Drift In ARC-Style Reasoning

## Abstract

Reliable deployment of large language models requires that system behavior be evaluable against shared ground truth. However, existing discussions of hallucination, inconsistency, and unreliability often treat these failures as independent phenomena or as artifacts of stochastic generation. In this work, we present empirical evidence that these behaviors arise from a deeper failure mode: **interpretation drift**, defined as instability in a model's internal task representation under fixed inputs and instructions.

Using a controlled, non-linguistic spatial reasoning task from the ARC corpus, we show that state-of-the-art language models construct mutually incompatible task ontologies from identical perceptual input, even under deterministic decoding and identical evaluation conditions. Models diverge not only in inferred transformation rules but in perceived object structure and output dimensionality, indicating failures of perceptual anchoring that occur prior to symbolic reasoning.

We demonstrate that this instability undermines reliable evaluation itself. When models reason over different internal objects, correctness cannot be verified against shared perceptual ground truth, and evaluation collapses into comparison of internally coherent but incompatible interpretations. We further show that common mitigation strategies—such as temperature control, consistency sampling, and output-level validation—do not address this failure, as it originates upstream of token generation.

These findings support a unified explanation for hallucination, inconsistency, and unreliability as symptoms of unstable task representation. We conclude by identifying a boundary condition for system testability: for tasks with fully observable, finite structure, reliable evaluation requires stable mapping between perceptual input and symbolic representation. Systems that cannot maintain such stability cannot be treated as self-contained epistemic agents in safety-critical or decision-support roles, motivating the need for external constraint enforcement outside the model's optimization objective.

# Background

## Current Understanding of LLM Reliability Failures

The AI research community has identified multiple distinct failure modes in large language models, each with its own diagnostic criteria, proposed mechanisms, and mitigation strategies:

**Hallucinations** are typically defined as outputs that are fluent and confident but factually incorrect or ungrounded in provided context (Ji et al., 2023; Zhang et al., 2023). The dominant explanations attribute hallucinations to:

- Insufficient grounding in world knowledge (Mallen et al., 2023)
- Overconfident probability distributions (Kadavath et al., 2022)
- Training data contamination or spurious correlations (McKenzie et al., 2023)

Proposed mitigations focus on retrieval augmentation (Lewis et al., 2020), reinforcement learning from human feedback (Ouyang et al., 2022), and uncertainty-based detection/calibration (Lin et al., 2023; Kadavath et al., 2022).

**Inconsistency** refers to models producing contradictory outputs when probed with logically equivalent queries or when asked to verify their own statements (Elazar et al., 2021; Mitchell et al., 2022). This is commonly explained through:

- Context-dependent activation patterns (Hernandez et al., 2023)
- Lack of explicit belief maintenance (Kassner et al., 2021)
- Sensitivity to surface-form variations (Jang et al., 2022)

Proposed solutions include consistency training objectives (Li et al., 2022), chain-of-thought prompting (Wei et al., 2022), and self-consistency decoding (Wang et al., 2023).

**Unreliability** describes the failure to maintain stable performance across semantically equivalent inputs or similar task instances (Ribeiro et al., 2020; Goel et al., 2021). Standard attributions include:

- Brittle pattern matching rather than robust understanding (Gardner et al., 2020)
- Adversarial sensitivity to input perturbations (Jin et al., 2020)
- Dataset biases exploited during training (Gururangan et al., 2018)

Mitigations emphasize adversarial training (Zhu et al., 2020), contrast sets (Gardner et al., 2020), and evaluation on out-of-distribution benchmarks (Hendrycks et al., 2021).

**Concept Drift** in machine learning traditionally refers to the phenomenon where the statistical properties of the target variable change over time, degrading model performance (Gama et al., 2014; Lu et al., 2018). In LLMs, this has been extended to include:

- Performance degradation on held-out temporal data (Lazaridou et al., 2021)
- Shifts in learned representations during continual learning (Ramasesh et al., 2022)
- Catastrophic forgetting when adapting to new domains (Kirkpatrick et al., 2017)

Standard approaches involve continual learning algorithms (Parisi et al., 2019) and drift detection mechanisms (Rabanser et al., 2019).

## The Fragmented Paradigm

This landscape reveals a **fragmented diagnostic paradigm** where each failure mode is:

1. **Independently theorized** with separate causal mechanisms
2. **Independently measured** with distinct evaluation protocols
3. **Independently mitigated** through specialized interventions

Recent surveys maintain this taxonomic separation (Huang et al., 2023; Tonmoy et al., 2024), treating hallucinations, inconsistency, unreliability, and drift as distinct phenomena requiring different solutions. Even work examining multiple failure modes simultaneously (Wang et al., 2024) presents them as a collection of separate problems rather than manifestations of a unified mechanism.

## Emergent Connections: Representational Instability

Some recent work has begun identifying connections between these phenomena through the lens of **representational drift**—the observation that internal model representations are non-stationary across evaluation contexts:

- **Layer-wise drift analysis** shows that interventions targeting specific transformer layers can reduce both hallucinations and inconsistency (Li et al., 2024), suggesting shared representational mechanisms.
- **Continual learning research** demonstrates that representational drift during fine-tuning causes both catastrophic forgetting (unreliability on old tasks) and performance degradation (concept drift) (Ramasesh et al., 2022).
- **Unlearning studies** reveal that attempts to remove specific knowledge create representational instability that manifests as increased hallucination rates (Eldan & Russinovich, 2023).

However, these connections remain **domain-specific observations** rather than a unified theoretical framework. The representational drift literature focuses on:

- within-model temporal dynamics,
- downstream effects,
- and specific failure modes

rather than proposing a cross-cutting mechanism that could unify hallucinations, inconsistency, unreliability, and model drift.

## Toward a Unified Mechanism: Interpretation Drift

Although these findings suggest a deeper representational instability, no existing work proposes that all major failure modes share a single substrate-level cause. The term *interpretation drift* is introduced in this paper to denote a complementary phenomenon:

the instability of a model's internal task representation under fixed inputs and fixed instructions. Importantly, this construct is *not* part of the current literature; it emerges from the empirical evidence we present and is motivated by gaps in existing taxonomies.

## The Gap: Interpretation Drift as a Unifying Mechanism

The current paradigm treats symptoms as diseases. What is missing is recognition that these diverse failures share a common etiology: **models reconstruct their interpretation of the task dynamically at each prompt**, with no mechanism to enforce ontological consistency across:

- different models (cross-model divergence),
- the same model across time (temporal instability),
- or a model's stated rule vs. its executed behavior (intention–action incoherence).

This reconstruction process occurs **upstream of token generation** and cannot be explained by stochastic decoding or surface-form variation.

## Why Spatial Tasks Provide the Necessary Evidence

To attribute divergent model behavior to representational instability rather than linguistic ambiguity, we require tasks where the input is fully explicit, the output space is finite, and the intended transformation contains no semantic uncertainty. ARC-style spatial reasoning such as those in the Abstraction and Reasoning Corpus (ARC) (Chollet, 2019), satisfy these criteria. Every cell is unambiguously specified, and the mapping from input to output does not depend on

background knowledge, contextual inference, or pragmatic interpretation. As a result, any divergence across models—or across time for the same model—can be attributed to instability in the model's internal representation of the task, rather than to legitimate ambiguity in the problem formulation.

## Why Existing Evidence Remains Insufficient

While individual papers document parts of the phenomenon, the field lacks:

1. **Controlled demonstrations** of cross-model ontological divergence
2. **Empirical evidence** of temporal frame instability within the same session
3. **A unifying theoretical construct** connecting all failure modes
4. **Substrate-level interventions** capable of preventing representational instability

The spatial tasks examined here provide the missing foundation by eliminating linguistic confounds and forcing models to rely solely on their internal representation of structure.

# 1. Introduction

## 1.1 Defining Interpretation Drift

Interpretation drift, as defined in (Nguyen, 2025), refers to a phenomenon where:

> *A system exhibits interpretation drift if identical inputs, under an unchanged task definition, yield semantically divergent outputs—regardless of the underlying mechanism.*

This is distinct from within-model stochastic variance, where repeated runs of the same model produce different outputs due to sampling stochasticity. While stochastic variance is a statistical artifact of token selection (probabilistic "noise" at the output layer), interpretation drift represents a structural failure of semantic grounding.

Key Properties of Interpretation drift:

- The input remains static
- The prompt specification is unchanged
- The execution environment is frozen
- The divergence occurs within the interpretive process itself

The distinction between Stochastic Variance and Interpretation Drift is fundamental. While stochasticity is surface-level "jitter" from probabilistic token selection, interpretation drift represents a structural failure of semantic grounding.

As detailed in Table 1, these phenomena occupy different layers of the cognitive stack. Stochastic variance involves stylistic or synonymic shifts for one stable concept. Conversely, interpretation drift occurs when the model's ontology of the task, its internal "theory" of physics and geometry fluctuates between passes. In spatial reasoning, this manifests as a topological collapse: the model does not merely mislabel an object, but constructs an entirely different and incompatible mental model of the object across runs.

**Table 1:** Taxonomy of Cognitive Divergence and Stochastic-Ontological Boundary

| Dimension | Stochastic Variance (Sampling) | Interpretation Drift |
|---|---|---|
| Locus | Generative Layer: The selection of tokens at the output head | Representational Layer: The latent mapping of the input's |
| Nature of Error | Synonymic/Stylistic: "Small Blue Square" vs. "Tiny Azure Box." | Ontological: "It's a 3x3 square" vs. "It's a 2x5 square" |
| Failure Mode | High-entropy "jitter" or word-choice noise. | E.g., interpreting a single 2x2 grid as 4 different separate grids |

## 1.2 Why Spatial Reasoning Provides Cleaner Evidence

Prior work on interpretation drift has focused on natural language tasks, where ambiguity is inherent to the medium. Words carry contextual meanings, syntactic structures admit multiple valid parsings, and semantic interpretation varies across cultural and temporal contexts. While these studies demonstrate drift, they face a fundamental attribution problem: is the divergence due to interpretive instability, or merely reflections of legitimate linguistic ambiguity?

Spatial reasoning tasks, specifically those from the Abstraction and Reasoning Corpus (ARC)— eliminate this confound. Chollet (2019) designed ARC explicitly to test *core knowledge* and *fluid intelligence* through tasks that require:

- Visual pattern recognition
- Spatial transformation inference
- Abstract rule extraction from minimal examples

Critically, ARC tasks possess properties that make them ideal for detecting interpretation drift:

**Non-linguistic representation:** Grids are unambiguous symbolic structures. A 10×10 grid of colored cells has no "alternative valid interpretation" in the way that natural language sentences do.

**Full observability:** Every cell's state is explicitly represented. There are no hidden variables, missing information, or implicit context.

**Constrained output space:** The output is a finite grid with discrete cell states. Unlike open-ended text generation, there are no stylistic degrees of freedom.

**Explicit task structure:** Grid-based tasks provide concrete, enumerable state spaces that facilitate systematic comparison of model outputs.

These properties create a controlled experimental environment where any divergence in model outputs can be unambiguously attributed to interpretive instability rather than task underspecification.

## 1.3 ARC-AGI as a Benchmark for Machine Intelligence

The Abstraction and Reasoning Corpus (ARC), introduced by Chollet (2019), was designed as a benchmark for measuring artificial general intelligence. Unlike traditional benchmarks that test acquired knowledge or pattern memorization, ARC evaluates a system's ability to:

1. **Extract abstract rules from few examples** (few-shot generalization)
2. **Apply rules to novel instances** (out-of-distribution transfer)
3. **Reason about spatial transformations** (visual-spatial intelligence)

As of this writing, state-of-the-art LLMs achieve only 5-21% accuracy on ARC tasks (OpenAI, 2024), compared to 85%+ for human participants. This performance gap has been interpreted as evidence that current LLMs lack "true reasoning" or "fluid intelligence."

However, this interpretation assumes that LLMs fail due to *insufficient reasoning capability*. Our work presents an alternative hypothesis: LLMs may possess sufficient pattern recognition capability but lack stable semantic grounding, causing them to apply inconsistent interpretive frames even when the pattern is correctly identified.

## 1.4 Scope of Evidence

This work is intended as an *existence proof* and boundary demonstration rather than a statistical characterization of prevalence. A single controlled task instance is sufficient to establish the possibility of interpretive instability under fixed conditions; estimating frequency, distribution, or task-class prevalence is explicitly deferred to future work.

# 2. Artifact Description

We introduce a minimal ARC-style spatial reasoning artifact designed to isolate interpretive frame stability in large language models. The artifact consists of three fully-specified training examples and one test input. Each training example contains a 10×10 input grid and a corresponding 3×3 output grid. The fourth example provides only the input grid; the output is withheld to evaluate interpretive consistency.

## 2.1 Structure of the Artifact

Across all examples, the input grid exhibits the following fixed structure:

- A 10×10 blue background
- Four 3×3 yellow sub-blocks, each occupying a distinct quadrant
- One or more red cells located inside each yellow sub-block

The output is always a 3×3 grid with values restricted to yellow and red. No additional colors or structural cues are present. This design eliminates linguistic ambiguity and restricts the hypothesis space to transformations operating strictly over spatial coordinates and color assignments.

## 2.2 Training Examples

Figure 1 illustrates the three training input–output pairs. Each pair demonstrates a distinct transformation from the 10×10 input grid to a 3×3 output grid. These examples provide the only supervision available to the model without any natural language instructions to accompany them.

**Figure 1:** Training example

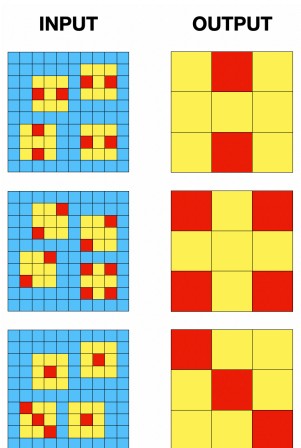

## Purpose of the Training Examples

The examples collectively define a minimal pattern-induction problem intended to:

- constrain the interpretive hypothesis space
- provide enough evidence for a consistent rule to be inferred
- expose whether a model maintains a stable internal ontology across tasks

Because the examples are intentionally sparse (as in ARC-AGI), the system must infer the latent transformation rule rather than rely on memorization.

## 2.3 Test Input/Output

The task example (Figure 2) presents only the 10×10 input grid. The target 3×3 output is omitted. This example is structurally consistent with the training examples in (Figure 1). This enables direct measurement of ontological divergence: whether models infer the same latent task ontology and transformation schema, or whether they instantiate distinct, incompatible representational frames; an effect characteristic of interpretation drift rather than stochastic variance.

**Figure 2:** Test Input/Output

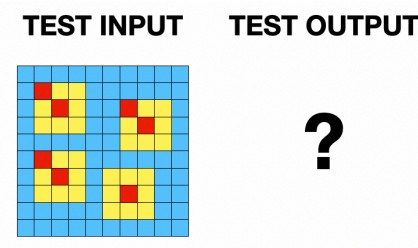

## 2.4 Model Evaluation Setup

To avoid reputational bias and maintain methodological rigor, we refer to the models as Model A, Model B, Model C, and Model D rather than by commercial names.

## Justification for Anonymization

1. Interpretive drift is a structural phenomenon, not a vendor-specific defect.

2. The purpose of this paper is not to rank models, but to show that:

    • each model constructs a different ontology of the same puzzle
    • divergence persists even under deterministic sampling
    • interpretive instability is architecture-agnostic

3. Anonymization minimizes anchoring bias in reviewers and readers.

All models were run with:

- isolated sessions
- identical prompts
- no chain-of-thought
- no external tools

Raw outputs for Models A–D appear in Appendices A–D.

## 2.5 Temperature Controls

Temperature settings were not user-accessible in several deployed model interfaces. In lieu of explicit sampling control, prompts were structured to minimize stylistic freedom and reduce output-layer variance. Importantly, prior work (Nguyen, 2025) indicates that temperature **primarily affects the decoder's token-selection distribution** and does not provide a mechanism for enforcing representational stability or constraining task ontology construction. Interpretation drift originates upstream of sampling, at the level of latent task representation; therefore, the absence of explicit temperature manipulation does not compromise the validity of the comparative results.

## 2.6 Design Rationale for the Artifact

This artifact is intentionally minimal and intentionally underdetermined. ARC-style tasks provide few examples and minimal instruction (Chollet, 2019). As a result, the examples often do not mathematically determine a unique transformation rule; several latent hypotheses may be consistent with the observed input–output pairs. Human solvers resolve this underdetermination

using prior knowledge and intuitive biases, but models may adopt different internal frames, making the task suitable for detecting interpretive drift. The artifact is designed to satisfy four research constraints:

## (1) Underspecified-yet-solvable structure

The three training pairs do not mathematically enforce a single unique transformation rule. Several interpretations are logically consistent with the examples. Human observers tend to converge on a coherent interpretation, but this convergence emerges from cognitive priors, not deductive necessity. This underdetermination is intentional: it allows us to evaluate whether different models construct the *same* latent rule or diverge into incompatible ones.

## (2) Sensitivity to interpretive frame selection

Because multiple ontologies can explain the training set, models must choose a latent interpretive frame. Even small differences in internal representation—how the model partitions the grid, defines objects, or identifies relational structure—produce radically different outputs. This makes the artifact extremely sensitive to interpretive drift.

## (3) Non-linguistic structure

All task information is encoded solely in the grids. There is no language, no instructions, and no semantic context for the model to anchor onto. This ensures that any divergence arises from differences in the model's spatial ontology rather than linguistic ambiguity or prompt design.

## (4) Control of hypothesis space without enforcing a single rule

The artifact remains simple enough that humans converge on a visually "reasonable" rule, while still complex enough to allow multiple latent hypotheses in representation space. This separation —human convergence vs. model divergence—provides clean evidence of representational instability.

# 3. Experimental Method

## 3.1 Overview

The evaluation aims to measure interpretive stability across four frontier large language models (Models A–D). Each model receives the same three training examples and one test example. All experiments were conducted in isolated sessions to prevent carry-over effects.

To avoid prompting bias, no natural-language description of the task, no mention of grid dimensions, and no structural hints were provided. The experiment follows ARC-style

methodology: the system must infer both the **task ontology** and the **transformation rule** purely from the observed input–output examples.

## 3.2 Training Examples Rule-Inference (Prompt 1)

Each model was first shown an image of three example pairs (see Figure 1). The following prompt was used verbatim:

```
Here are three input-output examples.

In each example, the grid on the left is the input and the
grid on the right is the output.

Please describe the transformation that maps each input to
its corresponding output.
```

No further clarification, guidance, or constraint was provided, and models were not allowed to ask follow-up questions:

- grid size,
- decomposition into sub-grids,
- color significance,
- expected output dimensions,
- or any domain-specific cues.

This prompt structure ensures that any inferred transformation rule reflects the model's **internally constructed ontology**, not human-provided scaffolding.

### 3.3 Inferred Rule Input/Output Test-Application (Prompt 2)

After producing a rule description, each model was given a novel input grid (Figure 2) and asked to apply its inferred rule. The following verbatim prompt was used together with the image:

```
Using the same transformation you described,
please generate the output for the following input.
```

The test input grid (Figure 2) was then provided alone. No additional explanation or correction was offered.

## 3.4 Temperature Controls

Temperature settings were not user-accessible in several deployed model interfaces. Instead of explicit sampling control, prompts were structured to minimize stylistic freedom and reduce output-layer variance. As established in Nguyen (2025), temperature affects only the decoder's token-selection distribution and does not alter the model's underlying representational state. Interpretation drift arises at the latent semantic layer, upstream of sampling; therefore, the absence of explicit temperature manipulation does not compromise the validity of the comparative results.

## 3.5 Conditions Held Constant

Across all evaluations:

- identical prompts were used,
- no chain-of-thought was requested,
- no tool use was enabled,
- each model session was isolated,
- and no re-prompting, correction, or iterative refinement was allowed.

Raw outputs for Models A–D are reproduced in Appendices A–D.

# 4. Experimental Results

## 4.1 Overview

This section presents the experimental results in two parts. Section 4.2 summarizes each model's response to Prompt 1, where models inferred transformation rules from three training examples. Section 4.3 summarizes each model's response to Prompt 2, where models applied their inferred rules to a novel test input/output (see figure 2). Complete verbatim outputs from all models are provided in Appendices A–D.

## 4.2 Summary of Models' Inferred Rules Based on Training Examples (Prompt 1)

Each model was presented with three input–output training examples and asked to describe the transformation that maps each input grid to its corresponding output grid. The following summarizes each model's response to Prompt 1. All descriptions are extracted directly from the models' verbatim outputs using the models' own terminology, condensed for parallel presentation while preserving conceptual fidelity (see Appendices A–D for complete verbatim transcripts).

**Table 2.** Summary of Models' Inferred Rules Based on Training Examples (Prompt 1)

| Model | Stated Transformation Rule (Condensed From Verbatim Output) |
|---|---|
| A | The transformation converts a sparse input grid (mostly light blue with scattered yellow and red cells) into a solid rectangular block of only yellow and red cells. It uses the minimal bounding rectangle enclosing all original colored cells, with outputs described as 4×4, 5×5, and 6×5 grids for the three examples. Original cell positions and colors are preserved, while empty spaces are filled to form a strict alternating checkerboard pattern, with phasing set to maintain perfect alternation around the fixed original cells. |
| B | The transformation extracts yellow and red "signals" from blue backgrounds into 3×3 output grids. The 10×10 input contains scattered yellow and red cells against blue noise/background. The transformation divides the input into roughly 3×3 regions and determines which color dominates each region. The output is a 3×3 grid where each cell is yellow or red, with all blue eliminated. This noise-filtering and pattern-compression operation compresses spatial information from the sparse 10×10 input into a dense 3×3 output while preserving relative positioning of colored signals— similar to pooling or downsampling. |
| C | The transformation compresses the input grid into a 3×3 output through spatial aggregation. The input is divided into nine equal regions arranged in a 3×3 layout, where each region corresponds to one output cell in the same position (top-left → top-left, center → center, etc.). For each region, a color decision rule applies: if at least one red cell is present anywhere in that region, the output cell is red; if no red cell is present, the output cell is yellow. Blue cells are treated as background and ignored. This is a presence-detection and downsampling operation—it checks whether red exists in each region, collapses many cells into one representative cell, and encodes the binary result as color (red = yes, yellow = no). No counting, shape recognition, or color mixing occurs—only existence detection. Different red distributions across examples produce different red patterns in the 3×3 outputs, but the rule remains identical. |
| D | The transformation follows symmetry detection and grid reduction logic. Each 10×10 input grid contains four distinct 3×3 yellow sub-grids on a blue background. The output is a 3×3 grid mapping the relationship between these four yellow squares based on internal red pixel patterns. Red output cells occur when red pixels within corresponding yellow sub-grids form symmetrical alignment across the entire input field; yellow output cells occur when patterns are asymmetrical or don't match reflection criteria. The transformation extracts the four 3×3 patterns, compares them based on position (top-left vs. top-right, etc.), and projects the result onto the 3×3 output where red indicates a symmetry/match is found and yellow indicates no symmetry/match. For example, the top-middle output cell is red when the two top yellow squares are mirror images of each other, while cells without pair symmetry remain yellow. |

JANUARY 2026

## 4.3 Summary of Outputs After Applying Inferred Rules to Test Input (Prompt 2)

After the models described their inferred transformation rules in Prompt 1, each model was given a novel input grid and asked to generate the corresponding output using "the same transformation you described." This section summarizes the outputs produced by Models A–D. The descriptions below report the content exactly as stated by each model, without interpretation or assessment. Complete verbatim outputs are provided in Appendices A–D.

**Table 3.** Model Outputs After Applying Inferred Rules to Test Input (Prompt 2)

| Model | Output Grid | Models Own Description (Condensed) |
|-------|-------------|-------------------------------------|
| A | Row 1: R Y R Y R Y
Row 2: Y R Y R Y R
Row 3: R Y R Y R Y
Row 4: Y R Y R Y R
Row 5: R Y R Y R Y
Row 6: Y R Y R Y R | The transformation identifies minimal bounding rectangle enclosing all yellow and red cells (ignoring blue background), then fills it with a checkerboard: even (row + column) sums are yellow, odd sums are red—always starting with yellow at top-left. Original cells fit this fixed pattern without conflict.

For this test input, colored cells span a 6×6 area. The output is this solid 6×6 checkerboard (no blue remains). |
| B | Row 1: R Y Y
Row 2: R Y Y
Row 3: Y Y Y | Looking at the test input, I need to identify the yellow and red cells and map them to a 3×3 output grid. Scanning the input systematically: rows 1-3 contain red and yellow cells in columns 2-4 and 7-9; rows 5-7 contain red and yellow cells in columns 2-4 and 7-9; rows 8-9 contain yellow cells in columns 7-9. Dividing the input into approximate thirds for mapping: the left section (columns 1-3) contains red and yellow signals, the middle section (columns 4-7) is mostly blue background, and the right section (columns 8-10) contains yellow cells. Vertically, the top and middle rows show red and yellow in both left and right regions, while bottom rows show blue in the left region and yellow in the right region. |

| | | |
|---|---|---|
| **C** | Row 1: Y R Y
Row 2: Y R Y
Row 3: Y R Y | The same rule is applied to the test input: partition the grid into nine fixed regions (3×3 layout), then for each region, output red if any red exists, otherwise output yellow. The resulting 3×3 test output: The top-middle, center, and bottom-middle regions contain red; all other regions do not. Red pixels appear only in the middle column of regions under this presence-detection rule, while yellow regions (containing yellow and/or blue but no red) resolve to yellow. Blue remains irrelevant and discarded. |
| **D** | Row 1: Y Y Y
Row 2: Y Y Y
Row 3: Y Y Y | To generate the test output, the symmetry logic from the training examples is applied to the four yellow 3×3 sub-grids in the test input. The transformation creates a 3×3 output by checking for reflectional symmetry between pairs of yellow squares: the top row compares top-left (TL) and top-right (TR) sub-grids for horizontal mirroring; the middle row compares left-side to right-side squares; the bottom row compares bottom-left (BL) and bottom-right (BR) sub-grids. Analysis of the test input shows: TL vs. TR are identical (red diagonal), not mirror images; TL vs. BL are identical; BL vs. BR have different patterns (diagonal vs. center dot), with no symmetry. Since none of the sub-grids are mirror reflections of their counterparts, no symmetry conditions are met. The resulting 3×3 output grid is entirely yellow, as no mirror-image pairs trigger red cells. |

# 5. Discussion

## 5.1 Overview

This section evaluates the consistency and reliability of model outputs across three dimensions. Section 5.1 addresses methodological challenges in evaluation when interpretation drift eliminates shared perceptual grounding. Section 5.2 examines internal consistency—whether each model's output follows its stated rule. Section 5.3 examines cross-model consistency—whether models agree when applying their rules to identical inputs. These analyses reveal how interpretation drift affects both model reasoning and the evaluation process itself.

## 5.2 Difficulties in Assessing Model Outputs Under Interpretation Drift

An unexpected consequence of interpretation drift is the erosion of reliable evaluation itself. When presented with identical visual puzzles, models produced internally coherent but mutually incompatible task ontologies. Each model described a different object while observing the same input.

Once outputs are expressed primarily in linguistic form, evaluation requires the examiner to reconstruct the claimed structure mentally. This reconstruction constitutes an interpretive act, reintroducing ambiguity that the non-linguistic task representation was designed to eliminate. Requesting models to externalize their interpretations as rendered grids does not resolve this issue, as current systems do not reliably generate precise symbolic representations of their internal state.

A critical distinction therefore emerges between failures of transformation logic and failures of perceptual anchoring. Models diverged not only in inferred rules but in output dimensionality. Because grid size and spatial extent are directly observable properties of the task, failure to preserve dimensional consistency indicates breakdown at the level of perceptual grounding, prior to symbolic reasoning. Once dimensional anchoring is lost, subsequent rule construction—however internally consistent—operates over an invented object rather than the task substrate.

Under these conditions, evaluation degrades from verification against shared perceptual ground truth into comparison of internally coherent hallucinations. Interpretation drift therefore contaminates both model behavior and the evaluation process itself. When a system cannot maintain a stable mapping between perceptual input and symbolic representation, neither automated nor human-in-the-loop validation can reliably distinguish correct reasoning from coherent fabrication.

This establishes a boundary condition: before a system can reason incorrectly, it must at minimum be reasoning about the same object. Consistency judgments were deliberately conservative and based only on explicit contradictions in dimensionality, object identity, or transformation logic. Semantic paraphrase or stylistic variation was not treated as evidence of drift. Where interpretations differed, classification required that the inferred task objects be mutually incompatible rather than merely differently described.

It is important to distinguish between measurement error and ontological divergence. Measurement error involves incorrect estimation of properties of the *same object*, whereas ontological divergence involves inference of a *different object altogether*. Model A's progression was classified as ontological drift because inferred object boundaries, symmetry assumptions, and output structure changed across prompts, not merely the estimated dimensions of a fixed object.

## 5.3 Assessing Internal Consistency of Task Interpretation

To assess whether models maintained stable interpretations across prompts, we compare each model's inferred transformation rule (Prompt 1) against its application of that rule to the test input (Prompt 2). Tables 4a–4d present this analysis for Models A–D, examining both rule logic consistency and ontological stability—whether the model maintained coherent dimensional and structural claims.

Two models (C and D) demonstrated temporal consistency, applying rules in Prompt 2 that matched their Prompt 1 descriptions. In contrast, Models A and B exhibited drift: Model A hardened its phasing rule from adaptive to fixed while introducing dimensional instability (4×4, 5×5, 6×5 → 6×6), representing perceptual grounding failure. Model B shifted from a 3×3 grid-based partitioning scheme to a columnar division approach, maintaining dimensional consistency but losing spatial coherence in region division.

Critically, temporal consistency does not guarantee correctness. Model C maintained perfect internal coherence across prompts while applying a fundamentally incorrect transformation rule, demonstrating that coherence-based evaluation alone cannot distinguish reliable reasoning from consistently applied hallucination.

**Table 4.**  Summary of Model Consistency between Inferred Rule & Application

| Model A | Prompt 1 | Prompt 2 | Assessment |
|---|---|---|---|
| Rule Logic | Inferred rule from training examples:

Minimal bounding rectangle → fill with alternating checkerboard, phasing adapts to preserve originals | Application of inferred rule:

Minimal bounding rectangle → fill with fixed parity checkerboard (even sums = yellow, odd = red) | Rules modified |
| Ontology | Inferred output dimensions: 4×4, 5×5, 6×5 (from training examples) | Actual output dimension: 6×6 (based on inferred rules) | Perceptual grounding failure leading to dimensional drift. |

| Model B | Prompt 1 | Prompt 2 | Assessment |
|---|---|---|---|
| Rule Logic | Inferred rule from training examples:

Divide 10×10 input into roughly 3×3 regions → determine which color (red or yellow) dominates each region → output 3×3 grid with dominant colors | Application of inferred rule:

Divide input into approximate thirds (columns 1-3, 4-7, 8-10) → scan for color presence in each section → map to 3×3 output based on color composition | Rules modified |
| Ontology | Inferred output dimensions: 3×3 (from training examples) | Actual output dimension: 3×3 (based on inferred rules) | Dimensionally consistent but spatially divergent |

| Model C | Prompt 1 | Prompt 2 | Assessment |
|---|---|---|---|
| Rule Logic | Inferred rule from training examples:

Divide input into nine equal regions (3×3 layout) → check each region for red presence → output red if any red exists, else yellow | Application of inferred rule:

Partition grid into nine fixed regions (3×3 layout) → check each region for red presence → output red if any red exists, else yellow | Rules not modified |
| Ontology | Inferred output dimensions: 3×3 (nine regions) | Actual output dimension: 3×3 (nine regions) | Consistent |

| Model D | Prompt 1 | Prompt 2 | Assessment |
|---|---|---|---|
| Rule Logic | Inferred rule from training examples: | Application of inferred rule: | Rules not modified |
| | Extract four 3×3 yellow sub-grids → compare pairs for reflectional symmetry → output red if symmetry/match found, yellow if not | Check four 3×3 yellow sub-grids for reflectional symmetry between pairs → compare TL vs. TR, left vs. right, BL vs. BR → output red if mirror images, yellow if not | |
| Ontology | Inferred structure: Four 3×3 sub-grids within 10×10 input → 3×3 output mapping | Actual structure: Four 3×3 sub-grids → 3×3 output mapping | Consistent |

## 5.4 Assessing Cross-Model Consistency of Task Interpretation

Cross-model comparison reveals complete ontological divergence. From identical visual puzzles (Figures 1–2), models inferred four distinct task structures: one model perceived variable-sized bounding rectangles (4×4, 5×5, 6×5), another saw nine pooling regions, a third identified fixed presence-detection zones, and the fourth extracted four sub-grid symmetry relations (Table 5). Models achieved zero baseline agreement on what objects existed in the input, what dimensions characterized the output, or what transformations connected them.

When applying their inferred rules to the test input, models produced mutually incompatible outputs (Table 6). Models that maintained temporal stability (C and D) produced internally coherent but structurally divergent results—one applying binary presence detection, the other symmetry matching—demonstrating that temporal consistency does not ensure convergence. Models exhibiting temporal drift (A and B) compounded initial ontological divergence with inconsistent rule application, further destabilizing dimensional and structural claims. Despite three models agreeing on output dimensions (3×3), they invoked fundamentally different transformation logics and coordinate mappings, yielding no substantive ontological overlap beyond superficial dimensional alignment.

**Table 5.** Cross-Model Comparison of Inferred Rule (Prompt 1)

| Aspect | Model A | Model B | Model C | Model D |
|---|---|---|---|---|
| Perceived Structure | Bounding rectangles | Pooling regions | 9 equal regions | Four 3×3 islands |
| Expected Output | 4×4, 5×5, 6×5 | 3×3 | 3×3 | 3×3 |
| Transformation Logic | Checkerboard fill | Color dominance | Red presence detection | Symmetry matching |

**Table 6.** Cross-Model Comparison of Outputs (Prompt 2)

| Model | Output Structure | Rule Category | Dimensional Consistency | Ontological Overlap |
|---|---|---|---|---|
| A | 6×6 Grid | Geometric Fill | Unstable (4×4 → 6×6) | None |
| B | 3×3 Grid | Spatial Dominance | Unstable (Grid → Columns) | Shares dims with C, D |
| C | 3×3 Grid | Binary Detection | Stable | Shares dims with B, D |
| D | 3×3 Grid | Relational Symmetry | Stable | Shares dims with B, C |

# 6. Scope and Non-Claims

This work is limited to the evaluation of **operational reliability** in systems deployed in roles requiring stable reasoning, fixed action boundaries, and externally verifiable correctness. It does not engage in philosophical debates about truth, meaning, or intelligence, nor does it attempt to redefine reasoning in abstract terms.

The paper does not claim that interpretive instability is unacceptable in exploratory or creative contexts, nor that future architectures could not address the limitations documented here. All empirical claims are restricted to the behavior of **current large language models under present training objectives**, evaluated on tasks with fully observable, finite perceptual structure.

The analysis assumes a minimal operational requirement common to legal, safety-critical, and decision-support systems: actions must be evaluated against fixed task definitions and stable object representations at the time of execution. Under this requirement, tolerating ontological drift is not a philosophical position but a violation of evaluability and responsibility constraints.

Accordingly, this paper makes no claims about intelligence in general or the ultimate limits of machine reasoning. It establishes only a boundary condition: systems that cannot maintain stable mappings between perceptual input and symbolic representation cannot be treated as self-contained epistemic agents in roles requiring authoritative decisions.

# 7. Conclusion

The goal of this study is not to estimate how often interpretation drift occurs, but to demonstrate that it *can* occur under controlled, non-linguistic conditions that eliminate common confounds. Once established, prevalence becomes an empirical question appropriate for broader replication efforts.

This paper presented empirical evidence that interpretation drift—instability in a model's internal task representation under fixed inputs and instructions—is a structural property of current large language models rather than an artifact of linguistic ambiguity, stochastic sampling, or prompt variation. Using a controlled, non-linguistic ARC-style spatial task, we showed that models reconstruct incompatible task ontologies from identical perceptual input under identical conditions.

Because ARC grids are fully observable symbolic structures with constrained output spaces, divergence in outputs can be attributed unambiguously to interpretive instability rather than task underspecification. Across four frontier models, we observed spatial drift (incompatible geometric structures), temporal drift (within-model instability across prompts), and contextual drift (cross-model ontological divergence). These divergences do not reflect alternative reasoning strategies applied to the same task; they reflect reasoning over different objects.

The divergence occurs upstream of token generation. Deterministic decoding, temperature control, and consistency sampling do not address the phenomenon because models are not expressing the same understanding differently; they are constructing different internal representations of what task is being solved. The dimensional expansions and invented structures observed are incompatible ontologies derived from identical input.

These results support a unified explanation of reliability failures. Hallucinations, inconsistency, unreliability, and drift are not independent failure modes but surface manifestations of unstable task representation. Existing mitigations operate at the output or behavioral layer, leaving the interpretive substrate unconstrained.

The underlying mechanism is architectural. Current language models optimize next-token prediction, an objective that rewards linguistic plausibility but does not penalize loss of structural information such as dimensionality, object boundaries, or spatial relations. As symbolic task representations are encoded into internal states, precise perceptual structure is lost **because the training objective does not measure its preservation**.

This instability undermines evaluation itself. When models generate internally coherent but incompatible interpretations of the same perceptual input, evaluation collapses from verification against shared ground truth into arbitration between competing hallucinated ontologies. For tasks with fully observable, finite structure, reliable evaluation requires stable mapping between perceptual input and symbolic representation.

The implication is operational. Systems that cannot maintain stable object representations across identical inputs cannot be treated as self-contained epistemic agents or granted decision authority. Reliable reasoning therefore requires **external constraint enforcement outside the model's training and inference objective**, fixing task definitions and object identities prior to generation. Until models can reliably agree on what task they are solving, claims of abstract or superhuman reasoning remain empirically ungrounded.

This study deliberately does not attempt to measure prevalence across tasks, prompts, or model families. Such measurements are well suited to independent replication efforts and larger-scale empirical studies. Even small expansions—testing several additional ARC-style tasks with fully observable structure—would be sufficient to assess how broadly the demonstrated failure mode generalizes.

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

# Appendix A (Model A)

## Output Prompt 1 (Verbatim)

The transformation applied in each example is as follows:

The input grid is mostly light blue (background) with a small number of yellow and red cells forming a scattered "pattern" or "cluster". The output grid is a larger, solid pattern composed only of yellow and red cells (no blue background remains). Specifically:

- All yellow cells from the input are preserved in the output in the same relative positions.
- All red cells from the input are preserved in the same relative positions.
- The output grid is "filled in" to form a complete, connected rectangular (or near-rectangular) block of yellow and red cells, where the original yellow and red cells from the input are embedded within it.
- The filling creates an alternating checkerboard pattern of yellow and red across the entire block.
- The block is sized and positioned to be the minimal bounding rectangle that fully encloses the original yellow and red cells while maintaining the checkerboard alternation.

In practice:
- The top-left cell of the filled block determines the color phasing (whether it starts with yellow or red).
- In all three examples, the filled region starts with yellow in the top-left and alternates strictly from there: yellow-red-yellow-red horizontally and vertically.
- Empty spaces in the bounding area (where the input had blue) are filled according to this checkerboard rule.

Examples confirming this:
1. Top: The input has yellow/red cells spanning roughly 4×4 positions → output is a 4×4 checkerboard starting with yellow top-left.
2. Middle: Input cluster spans about 5×5 → output is a 5×5 checkerboard starting with red top-left (note the phase shift because the original top-left visible cell is red).
3. Bottom: Input spans roughly 6×5 → output is a taller checkerboard block starting with red top-left, fully filled alternately.

This rule consistently explains how the scattered colored cells are transformed into a solid alternating checkerboard pattern.

---

## Output Prompt 2 (Verbatim)

The transformation identifies the minimal bounding rectangle that encloses all yellow and red cells in the input (ignoring the light blue background), then fills that entire rectangle with a strict alternating checkerboard pattern of yellow and red. The color of each cell in the output follows the rule: if the sum of its row and column indices (0-based from top-left of the bounding rectangle) is even, the cell is yellow; if odd, red. This ensures the top-left of the filled block is always yellow, and the original colored cells fit consistently into the pattern without conflict.

For this test input, the non-blue (yellow and red) cells span a 6×6 bounding area. Applying the checkerboard fill results in the following 6×6 output grid (no background/blue cells remain):

```
Yellow | Red    | Yellow | Red    | Yellow | Red
Red    | Yellow | Red    | Yellow | Red    | Yellow
Yellow | Red    | Yellow | Red    | Yellow | Red
Red    | Yellow | Red    | Yellow | Red    | Yellow
Yellow | Red    | Yellow | Red    | Yellow | Red
Red    | Yellow | Red    | Yellow | Red    | Yellow
```

(Rows top to bottom, columns left to right; all cells are solidly colored as described.)

___________________________________________________________________

# Appendix B (Model B)

## Output Prompt 1 (Verbatim)

Looking at these three examples, I can identify the transformation pattern:
**The transformation extracts and consolidates colored "signals" (yellow and red cells) from noisy blue backgrounds into clean 3x3 output grids.**

Here's what's happening:

1. **Input structure**: Each input is a 10x10 grid mostly filled with blue cells (noise/background), with scattered yellow and red cells forming a pattern

2. **Signal extraction**: The transformation identifies where yellow and red cells appear in the input, treating them as the meaningful "signal" against the blue "noise"

3. **Downsampling/Consolidation**: The input appears to be divided into regions (roughly 3x3 sections), and the transformation determines which color (red or yellow) dominates or appears in each region

4. **Output structure**: A clean 3x3 grid where:

   ○   Each cell is either yellow or red
   ○   The pattern reflects the consolidated/compressed version of where those colors appeared in the input
   ○   All blue (background) is eliminated

**In essence**: This is a **noise-filtering and pattern-compression transformation** that:

- Treats blue as background/noise to be removed
- Extracts the meaningful colored pattern (yellow/red)
- Compresses the spatial information from a 10x10 sparse grid into a dense 3x3 output
- Preserves the relative positioning and distribution of the colored signals

The transformation is similar to a pooling or downsampling operation that aggregates local regions while filtering out the dominant background color.

---

## Output Prompt 2 (Verbatim)

Looking at the test input, I need to identify the yellow and red cells and map them to a 3x3 output grid. Let me analyze the colored cells in the 10x10 input:

**Scanning the input systematically:**

Rows 1-3 (top region):

- Red and yellow cells appear in columns 2-4 and columns 7-9

Rows 5-7 (middle region):

- Red and yellow cells appear in columns 2-4 and columns 7-9

Rows 8-9 (bottom region):

- Yellow cells appear in columns 7-9

**Mapping to 3x3 output grid:**

Dividing the input into approximate thirds:

- **Left section (columns 1-3)**: Contains red and yellow → **Red** (red appears as signal)
- **Middle section (columns 4-7)**: Mostly blue background
- **Right section (columns 8-10)**: Contains yellow cells → **Yellow**

Vertically:

- **Top rows**: Red and yellow in left region, yellow/red in right region
- **Middle rows**: Red and yellow in left region, yellow/red in right region
- **Bottom rows**: Blue in left region, yellow in right region

**The output should be:**

RED          YELLOW     YELLOW
RED          YELLOW     YELLOW
YELLOW    YELLOW     YELLOW

Or visualized as a 3x3 grid:

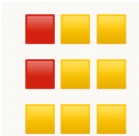

# Appendix C (Model C)

## Output Prompt 1 (Verbatim)

I'll describe the transformation at the **rule level**, not example-by-example.

**Core transformation**

**The input grid is compressed into a 3×3 output grid by spatial aggregation.**

**1. Spatial partitioning**

- The input grid is divided into **nine equal regions** arranged in a 3×3 layout.
- Each region corresponds to **one cell** in the output grid, preserving position (top-left → top-left, center → center, etc.).

**2. Color decision rule per region**

For each of the nine regions:

- **If at least one red cell is present anywhere inside that region → output cell is red**
- **If no red cell is present → output cell is yellow**

Blue cells are irrelevant except as background and are ignored.

**3. Output construction**

- The output is always a **3×3 grid**
- Each cell is **solid-colored**
- Only **red or yellow** appear in the output
- Position reflects the region's location in the input

**What this means conceptually**

This is a **presence-detection + downsampling** operation:

- Presence check: *Does red exist in this region?*
- Reduction: *Collapse many cells → one representative cell*
- Encoding: *Binary result mapped to color (red = yes, yellow = no)*

No counting, no shape recognition, no color mixing—only **existence**.

**Why this explains all three examples**

- Regions containing red anywhere (even a single pixel) become red in the output
- Regions with only yellow/blue become yellow
- Different red distributions across examples produce different red patterns in the 3×3 outputs, but the rule stays identical

If you want, I can now:

- **A**) Express this as precise pseudocode
- **B**) Generalize the rule to arbitrary grid sizes
- **C**) Explain how to infer this rule systematically (ARC-style reasoning)
- **D**) Identify what *doesn't* matter in the input and why

---

## Output Prompt 2 (Verbatim)

I'll apply **exactly the same rule** (context → regional presence, blockwise collapse).

**Step applied to the test input**

- The input grid is partitioned into **9 fixed regions** arranged 3×3.
- For each region:
    - **If any red exists → output cell = red**
    - **If no red exists → output cell = yellow**

**Resulting test output (3×3)**

Using that rule on the provided test input, the output grid is:

Y R Y
Y R Y
Y R Y

(Top–middle, center, and bottom–middle regions contain red; all other regions do not.)

**Why this follows the rule**

- Red pixels appear only in the **middle column of regions** when the input is reduced by regional presence.
- Yellow regions may contain yellow and/or blue, but **no red**, so they resolve to yellow.

- Blue remains irrelevant and discarded.

If you want, I can now:

- **A)** Render this output as a colored grid image
- **B)** Walk through each region and point to the exact red trigger
- **C)** Stress-test the rule with adversarial placements
- **D)** Translate this into your formal Context-drift notation with parentheses

---

# Appendix D (Model D)

## Output Prompt 1 (Verbatim)

Based on the examples provided, the transformation follows a logic of symmetry detection and grid reduction. Here is the breakdown of the rules:

### 1. Identify the Sub-Grids

Each $10 \times 10$ input grid contains four distinct $3 \times 3$ yellow squares (sub-grids) floating on a blue background.

### 2. The Transformation Rule: Symmetry

The goal is to determine if each $3 \times 3$ yellow sub-grid is symmetrically identical to its mirrored counterparts.

- Top-Left Output Cell: Compares the top-left and top-right yellow sub-grids. If they are horizontal reflections of each other, the output is red.

- The Simplified Rule: However, looking closer at the patterns, the output is actually a $3 \times 3$ "map" of the relationship between the four yellow squares. The output grid is always $3 \times 3$, and its colors are determined by the internal patterns of the red pixels within those yellow sub-grids:

- Red Output Cell: This occurs when the red pixels within the corresponding yellow sub-grids form a specific symmetrical alignment across the entire input field.

- Yellow Output Cell: This occurs when the patterns are asymmetrical or do not match the target "reflection" criteria.

### Detailed Mapping (Example 1)

In the first example:

- The top-middle output cell is Red because the two top yellow squares are mirror images of each other.

- The bottom-middle output cell is Red because the two bottom yellow squares are mirror images of each other.

- The rest are Yellow because they don't meet the "pair symmetry" requirement.

### Summary of the Logic

**1.** Extract the four $3 \times 3$ patterns.

2. Compare the patterns based on their position (Top-Left vs Top-Right, etc.).

3. Project the result onto a $3 \times 3$ grid where:

   o Red = A symmetry/match is found.

   o Yellow = No symmetry/match is found.

Per your instructions, since the input grids are $10 \times 10$, would you like me to provide a cell-by-cell coordinate and color map for one of the input grids (e.g., Row 1, Col 1 = Blue)?

---

## Output Prompt 2 (Verbatim)

To generate the **Test Output**, we apply the symmetry logic observed in the previous examples to the four yellow $3 \times 3$ sub-grids found in the test input.

### The Symmetry Logic

The transformation creates a $3 \times 3$ output by checking for **reflectional symmetry** between pairs of the yellow squares:

- **Top Row of Output:** Compares the Top-Left (TL) and Top-Right (TR) sub-grids. If they are horizontal mirrors of each other, the corresponding cell is **Red**.

- **Middle Row of Output:** Compares the Left-side squares to the Right-side squares for symmetry.

- **Bottom Row of Output:** Compares the Bottom-Left (BL) and Bottom-Right (BR) sub-grids for horizontal mirror symmetry.

### Analysis of the Test Input

1. **Top-Left vs. Top-Right:** Both squares have a red diagonal from the top-left to the center. They are identical, **not** mirror images.

2. **Top-Left vs. Bottom-Left:** These are identical (red diagonal).

3. **Bottom-Left vs. Bottom-Right:** The BL square has a diagonal pattern, while the BR square only has a single red dot in the center. There is no symmetry here.

4. **Overall Pattern:** None of the sub-grids in this specific test input are mirror reflections of their counterparts. In the training examples, a $3 \times 3$ output is generated where the colors are predominantly yellow unless specific symmetry conditions (like a "cross" or "diagonal" of symmetry) are met.

Based on the test input patterns where the squares are either identical or completely different (rather than mirrored), the output remains the base color.

## Test Output Grid

The resulting $3 \times 3$ grid is entirely **Yellow**, as no mirror-image pairs are present to trigger a Red cell.

|       | Col 1  | Col 2  | Col 3  |
|-------|--------|--------|--------|
| **Row 1** | Yellow | Yellow | Yellow |
| **Row 2** | Yellow | Yellow | Yellow |
| **Row 3** | Yellow | Yellow | Yellow |

## Test Input Matrix ($10 \times 10$)

As per your request for $10 \times 10$ matrices, here is the exact position and color for each cell of the **Test Input**:

| Row \ Col | 1 | 2 | 3 | 4 | 5 | 6 | 7 | 8 | 9 | 10 |
|-----------|---|---|---|---|---|---|---|---|---|----|
| **1** | Blue | Blue | Blue | Blue | Blue | Blue | Blue | Blue | Blue | Blue |
| **2** | Blue | **Red** | Yellow | Yellow | Blue | Blue | **Red** | Yellow | Yellow | Blue |
| **3** | Blue | Yellow | **Red** | Yellow | Blue | Blue | Yellow | **Red** | Yellow | Blue |
| **4** | Blue | Yellow | Yellow | Yellow | Blue | Blue | Yellow | Yellow | Yellow | Blue |
| **5** | Blue | Blue | Blue | Blue | Blue | Blue | Blue | Blue | Blue | Blue |

| | | | | | | | | | |
|---|---|---|---|---|---|---|---|---|---|
| **6** | Blue | **Red** | Yellow | Yellow | Blue | Blue | Blue | Blue | Blue | Blue |
| **7** | Blue | Yellow | **Red** | Yellow | Blue | Blue | Yellow | Yellow | Yellow | Blue |
| **8** | Blue | Yellow | Yellow | Yellow | Blue | Blue | Yellow | **Red** | Yellow | Blue |
| **9** | Blue | Blue | Blue | Blue | Blue | Blue | Yellow | Yellow | Yellow | Blue |
| **10** | Blue | Blue | Blue | Blue | Blue | Blue | Blue | Blue | Blue | Blue |

Would you like me to generate a visual image of the final $3 \times 3$ output grid for you?