# OpenReview forum: "Position: Interpretation Drift Is a Distinct Source of Instability in Large Language Models"
_ICML.cc/2026/Position_Paper_Track — Submitted to ICML 2026 Position Paper Track_

### Official Review · Reviewer_E95t · 2026-03-01

**Significance:** 1
**Argument Clarity:** 1
**Rating:** 2
**Confidence:** 4

**Questions:**

1. Why is interpretation drift a bad thing; is it not the case that this phenomenon also exists with humans?

**Alternative Views Section:**

Yes

**Compliance With Llm Reviewing Policy A Conservative:**

Affirmed.

**Discussion Potential:**

2

**Final Justification:**

As mentioned in the rebuttal acknowledgment, it is still not clear why interpretation drift is an important phenomenon or whether it is not just a natural phenomenon in LLMs. As a result, I maintain my original judgement.

**Paper Summary:**

The authors put forth the position that interpretation drift is a large source of instability between different models. They argue that LLMs lack of consistency in explanations or interpretations, even when holding the prompt constant and setting the temperature parameter to zero. The bulk of their argument rests on a set of three experiments where an LLM is asked to interpret an inherently ambiguous situation. Their results demonstrate that LLMs divergence in interpretation even in very controlled settings.

**Position:**

Yes

**Position In Title:**

Yes

**Related Work:**

2

**Strengths And Weaknesses:**

## Strengths
1. **Empirical Study** - The authors run an empirical study by looking at three sets of prompts. The results of the study demonstrate that interpretations of prompts vary from language model to language model, thereby proving one of the authors main claims. The details of the study are clearly presented in the paper, including the prompts used and the output for each model.

## Weaknesses
1. **Unclear Why Interpretation Drift is Important** - The central tenant of the paper is that interpretation drift is an important phenomenon that exists due to inconsistencies between language models. Yet after reading the paper, it is unclear to me why a) such a phenomenon is important, and b) why we should try and reduce or tackle this issue. At a high level, the authors demonstrate that language models which differ in their model, training data, and training procedure produce different interpretations for highly subjective prompts. This by itself is not surprising, and similar phenomena would be found if you asked humans the same question. It's unclear why we expect different language models to output the same response or explanation in the presence of ambiguity, and this point is little explored.
2. **Ambiguous or Made Up Tasks** - The authors propose three tasks to measure interpretation incoherence, but each of the three tasks little mimic real-world settings due to the lack of clarity in the directions. For example, Experiment A involves classifying products based on a prompt about identifying a product from their price. Even after reading the task several times, I'm not sure how to solve it, and moreover, it's not clear to me what types of model abilities this task aims to reflect. Experiment B is the most realistic, as it asks LLMs to deal with uncertainty; this task I think is ok, and at least somewhat seems to prove their point. Experiment C involves essentially nonsense, and so is a very out-of-distribution prompt where I would almost expect incoherence between language models. There's no inherent reason that I would want language models to have similar outputs when faced with significantly out-of-distribution language, and the authors fail to explain why such an experiment is needed.
3. **Measurability of Interpretation Drift** - In the introduction, the authors propose a way to measure interpretation drift, but yet no measurement paradigm is proposed. Rather, measurement is done in an ad-hoc way based on the outputs from the different experiments. It's unclear how such a qualitative and subjective evaluation can be scaled or applied across other language models to better understand their trends. If it was clear a) why interpretation drift is important, and b) how to measure it, then I think it would be a valuable tool for future research. But as it stands now, neither claim is quite clear, making it hard to understand where this stands in the relevant literature.

**Support:**

2

---

> ### Author Rebuttal · Authors · 2026-03-24
>
> Thanks for the feedback — we clarify why this phenomenon matters, the role of the experiments, and how it can be measured.
>
> Why interpretation drift matters.
> We agree that different models producing different answers is not surprising in itself, and that humans also disagree under ambiguity.
>
> The issue is not disagreement per se, but where it appears in the system.
>
> In many applications, LLM outputs are used as inputs to downstream logic (e.g., routing, classification, decision rules). These systems assume that the same input produces a stable interpretation. If two models produce different interpretations of the same input, the system may take different actions (e.g., approve vs reject, route A vs route B) without any change in the data.
>
> Human disagreement is usually visible and contextualized. In LLM systems, this divergence is silent: outputs are fluent and appear valid, but encode different decision logic.
>
> The concern is therefore operational: identical inputs can lead to different system behavior depending on which model is used.
>
> On ambiguity and “made-up” tasks.
> We agree that some tasks are ambiguous. This is intentional.
>
> The goal is to show that when inputs allow multiple reasonable interpretations, models select different ones. This situation is common in practice (e.g., classification with incomplete criteria, decision-making under partial information).
>
> Experiment A: demonstrates divergence in a structured but under-specified classification setting.
> Experiment B: shows the same effect in a realistic business decision task.
> Experiment C: is a control case — when no constraints exist, models still produce confident but incompatible outputs.
>
> The point of Experiment C is not realism, but calibration: it shows that coherence alone is not evidence of shared interpretation.
>
> On measurability.
> We agree that the current evaluation is qualitative.
>
> A more systematic measurement would involve:
>
> defining output categories (e.g., decision class, risk level),
> measuring agreement rates across models for identical inputs,
> and tracking how often models produce incompatible outputs.
>
> The paper focuses on demonstrating that the phenomenon is observable; formal metrics are a natural next step.
>
> On “why should models agree?”
> The claim is not that models should always agree.
>
> The issue is that many systems implicitly assume they do. If different models (or model versions) produce different interpretations, system behavior becomes unstable even when inputs are unchanged.
>
> This matters when outputs are used for decisions, not just text generation.
>
> Author note. The author is a non-native English speaker and used LLM-based tools to assist with language clarity and editing. All ideas, claims, and conclusions in this work are the author’s own.

---

> > ### Author Rebuttal · Reviewer_E95t · 2026-04-01
> >
> > Even after reading the rebuttal, I'm not sure if I understand the position taken by the authors. For example, the sentence "The claim is not that models should always agree. The issue is that many systems implicitly assume they do." does not really mean much when I read it. Overall, I think the importance of interpretation drift is not well-defended by the authors in the rebuttal.

---

> > > ### Author Response · Authors · 2026-04-01
> > >
> > > Thank you once more.
> > >
> > > The concern is not that models disagree. It is that downstream systems assume they don’t.
> > >
> > > When an LLM output feeds routing logic, a classification system, or a decision rule, that system treats the output as a stable signal. If two model versions produce different interpretations of identical inputs, the system takes different actions — approve vs. reject, route A vs. route B — with no change in the underlying data.
> > > Human disagreement is visible and contextualized. LLM divergence is silent: both outputs are fluent, confident, and appear valid. The system cannot detect it. This is the failure mode.
> > >
> > > On task design:
> > > Ambiguity is the point. Experiment C is not testing realism — it is a calibration case demonstrating that fluent, confident outputs are not evidence of shared interpretation. Coherence ≠ agreement.
> > >
> > > On measurability:
> > > This paper demonstrates the phenomenon is observable. It’s a ”proof of existence” and interpretation drift ”establishing” piece.
> > >
> > > What you are asking for will be in published   concurrently. My next paper ”Substrates are all we need: Defeating non-determinism with natural language” provide the measurement mechanism: the collapse the interpretation space to a singleton, then measure cross-model convergence via SHA-256 hash agreement. That is a formal, binary, scalable metric.
> > >
> > > With deterministic interpretation (opposite of interpretation drift) I empirically demostrate 4 frontier model produce byte for byte identical output for the same input.
> > >
> > > This will be the rigor the scientific community need for falsifiability. LLMs cannot be falsified today. But they will be with cross-model convergence. Same input, same output of identical SHA hashes offer a stable baseline.
> > >
> > > You might be familiar with Thinking Labs, trying to solve ”non-determinism”. Cross-model convergence is interpretive determinism and therefore be the an important finding in the next phase of AI. Moving from non-determinism to classical software behavior. This piece explaina the importance and relevant of determinism.
> > >
> > > https://hackernoon.com/re-defeating-nondeterminism-in-llm-inference-the-future-is-predictable

---

### Official Review · Reviewer_Rxts · 2026-03-09

**Significance:** 2
**Argument Clarity:** 2
**Rating:** 2
**Confidence:** 3

**Questions:**

1. How do you formally define interpretation drift?
2. What is the difference compared to the related work that studies the consistency metrics and mitigation methods?
3. Could the authors give insights on how to mitigate or address this issue?

**Alternative Views Section:**

Yes

**Compliance With Llm Reviewing Policy A Conservative:**

Affirmed.

**Discussion Potential:**

2

**Paper Summary:**

This paper empirically demonstrates the phenomenon of interpretation drift in large language models and argues that such drift can negatively impact the stability of real-world LLM-based applications. The authors illustrate this issue through a set of empirical examples.

**Position:**

Yes

**Position In Title:**

Yes

**Related Work:**

1

**Strengths And Weaknesses:**

Strengths:

The paper addresses an important and practical problem—interpretation consistency—in real-world deployments of LLM applications.

Weaknesses:

1. Interpretation consistency is not a new problem in the LLM applications. There exists many prior work that studies consistency metrics, and mitigation strategies for inconsistent model behaviors. The paper does not clearly position its contributions relative to these existing studies, nor does it articulate what is fundamentally new compared to prior work.
2. The paper primarily focuses on empirical demonstrations of interpretation drift, but lacks a deeper analysis of the underlying causes, principled measurement frameworks, or actionable insights on how to mitigate or address this issue. As a result, the current contribution remains largely observational.

**Support:**

2

---

> ### Author Rebuttal · Authors · 2026-03-24
>
> Thank you for the feedback. We address the concerns on definition, positioning, and scope directly.
>
> On formal definition.
> Interpretation drift is defined here at the input-output level:
>
> Given identical input
> 𝐷
> D and task definition
> 𝑇
> T, a set of models
> {
> 𝑀
> 𝑖
> }
> {M
> i
> 	​
>
> } exhibits interpretation drift if the resulting outputs are semantically incompatible (e.g., different decision classes, criteria, or framing), despite each output being internally coherent.
>
> The key properties are:
>
> input and task definition are fixed,
> divergence is measured at the semantic level (not token-level variation),
> and the phenomenon is defined independent of internal model mechanisms.
>
> We will revise the paper to make this definition explicit and separate it from causal explanations.
>
> On relation to prior work on consistency.
> We agree that consistency has been studied extensively. However, most prior work focuses on:
>
> within-model consistency (e.g., sampling variance, reasoning stability),
> or format/label consistency under fixed evaluation schemes.
>
> This paper focuses on a different setting:
>
> cross-model divergence under identical inputs and task definitions,
> where each model produces a single stable output, but the outputs reflect different implicit interpretations.
>
> In other words:
>
> prior work measures whether a model is consistent with itself,
> this work examines whether different models are consistent with each other under the same task.
>
> We will revise the related work section to make this distinction explicit and avoid overlap with existing consistency literature.
>
> On novelty.
> The contribution is not the general observation that models can be inconsistent, but the identification of a specific regime where:
>
> inputs and task definitions are unchanged,
> individual model outputs are internally stable,
> yet cross-model outputs are systematically incompatible.
>
> This isolates a failure mode that is not addressed by methods targeting within-model variability (e.g., re-sampling or aggregation).
>
> On “observational” nature of the contribution.
> We agree that the current paper is primarily empirical and diagnostic. This is intentional. The goal is to establish that the phenomenon exists and is observable under realistic conditions before proposing formal frameworks or mitigation strategies.
>
> We will revise the paper to make this scope explicit.
>
> On measurement and analysis.
> We agree that a principled measurement framework is needed. A natural extension would include:
>
> defining semantic equivalence classes for outputs (e.g., decision class, risk level, or framing category),
> measuring cross-model agreement rates under fixed inputs,
> and evaluating how agreement changes under controlled prompt constraints.
>
> We will clarify this direction in the paper.
>
> On mitigation.
> This paper intentionally does not propose mitigation strategies. The premise is that mitigation requires first isolating the failure mode.
>
> At a high level, mitigation would require:
>
> constraining the admissible interpretation space (e.g., via explicit task specification), or
> introducing external mechanisms to enforce alignment of interpretation across models.
>
> We will add a brief discussion clarifying that mitigation is out of scope, but follows from making interpretation explicit.
>
> Author note.
> The author is a non-native English speaker and used LLM-based tools to assist with language clarity and editing. All ideas, claims, and conclusions in this work are the author’s own.

---

> > ### Author Rebuttal · Reviewer_Rxts · 2026-04-03
> >
> > The authors' said "This paper focuses on a different setting that inputs and task definitions are unchanged, individual model outputs are internally stable, yet cross-model outputs are systematically incompatible". However, this observation is not new. Moreover, I cannot learn new insights since "This paper intentionally does not propose mitigation strategies."

---

> > > ### Author Response · Authors · 2026-04-03
> > >
> > > Thank you,
> > >
> > > This paper only establishes proof of existence of interpretation drift. What is new is ontological reconstruction of task. This paper proves that LLMs do not solve the same task and re-interpret the task and solves the exact same task with a different ontology each time. Whether this is important or not is argued in my next paper.
> > >
> > > The formal definitions and mitigation is also provided in my next paper (unsuitable as position paper) ”Substrates are all you need: Defeating non-determinism with natural language”
> > >
> > > The abstract here answers all your questions regarding? Formalism, mitigation/ solution to interpretation drift and why current evaluations do not surface this upstream failure mode.
> > >
> > >
> > > ABSTRACT
> > >
> > > Large language models (LLMs) exhibit output variance even under determinis-
> > > tic decoding (τ = 0), a phenomenon typically attributed to sampling artifacts or
> > > implementation details. We argue this framing is incomplete and identify an up-
> > > stream source of instability: interpretation drift, a condition in which multiple se-
> > > mantically valid but mutually incompatible task interpretations remain admissible
> > > under the same input (|I(D)|> 1). Independently trained models then produce
> > > divergent yet internally coherent outputs for identical prompts. Prompt engineer-
> > > ing, temperature-zero decoding, and constrained output formats (e.g., JSON mode
> > > or schema enforcement) all operate within the interpretation space: they bias se-
> > > lection but cannot collapse |I(D)|to a singleton. Substrates, by contrast, operate
> > > on the interpretation space itself.
> > > We introduce substrate-first architectures, in which task-specific sub-
> > > strates—external constraint specifications—reduce the admissible interpretation
> > > space to a singleton prior to token generation (|IS (D)| = 1). Under this
> > > condition, independently trained frontier models converge to byte-identical
> > > outputs at the decision surface, exhibiting behavior analogous to conventional
> > > deterministic software. Validation across Grok, GPT, Claude, and Gemini on
> > > deliberately ambiguous classification tasks shows cross-model agreement rising
> > > from ≈30% to 100%, with all models producing identical outputs verified by
> > > matching SHA-256 hashes (probability ≈ 2−768 under the null hypothesis of
> > > independent random generation).
> > > These results establish interpretation and computation as distinct architectural lay-
> > > ers, requiring a foundational shift from post-generation filtering to pre-generation
> > > constraint. Determinism in LLM systems is therefore not a property of the decod-
> > > ing process, but of whether the interpretation space is uniquely specified prior to
> > > execution. By externalizing interpretation prior to generation, we shift the locus
> > > of reliability from model behavior and post-hoc filtering to specification design.
> > > Under substrate-first architectures, outputs become reproducible, falsifiable, and
> > > auditable. This removes a fundamental barrier to high-stakes deployment and
> > > establishes the foundation for deterministic multi-agent systems. Reliable AI sys-
> > > tems are not achieved by fixing the model, but by fixing interpretation.

---

### Official Review · Reviewer_F1iR · 2026-03-13

**Significance:** 2
**Argument Clarity:** 3
**Rating:** 3
**Confidence:** 4

**Questions:**

- In the prediction, what kind of decoding did you use?
- Inconsistent behavior of models has already been reported in [1], and even its solution has been proposed. Based on this situation, how do you clarify the position of your paper?

[1] Xuezhi Wang, Jason Wei, Dale Schuurmans, Quoc V Le, Ed H. Chi, SHARAN NARANG, Aakanksha Chowdhery, Denny Zhou, "Self-Consistency Improves Chain of Thought Reasoning in Language Models", In ICLR 2023.

**Alternative Views Section:**

Yes

**Compliance With Llm Reviewing Policy A Conservative:**

Affirmed.

**Discussion Potential:**

2

**Final Justification:**

After the rebuttal, I understand what kinds of models are used. However, the details of the used models are still unclear. At least the authors should have shared the used temperature. Regarding the position of Self-consistency, the authors clarified its position from the paper. Due to this reason, I've decided to increase my score.

**Paper Summary:**

In this paper, the authors focus on the interpretation drift, which contradicts the commonly believed characteristic of models, semantic invariance, which gives the same interpretation across models, runs, and time. To reveal the existence of interpretation drift, the authors investigated anonymized models in three experiments, go-to-market (GTM) classification, estimating M&A valuation, and answering nonsense queries. Experimental results show that interpretation drifts exist in all settings.

**Position:**

Yes

**Position In Title:**

Yes

**Related Work:**

1

**Strengths And Weaknesses:**

Strengths:
- The interpretation drift investigated by this paper is an important aspect and deficiency of large language models (LLMs).
- The experiment is conducted on three different tasks.
- Qualitative analysis is conducted.

Weaknesses:
- Since the model names are anonymized, readers cannot judge that the results are generalizable to other cases.
- As shown in the paper of Self-Consistency [1], semantic inconsistency has already been observed, and its solution is also proposed. Thus, this viewpoint itself is not novel.

[1] Xuezhi Wang, Jason Wei, Dale Schuurmans, Quoc V Le, Ed H. Chi, SHARAN NARANG, Aakanksha Chowdhery, Denny Zhou, "Self-Consistency Improves Chain of Thought Reasoning in Language Models", In ICLR 2023.

**Support:**

2

---

> ### Author Rebuttal · Authors · 2026-03-24
>
> Thanks for the feedback — we clarify the positioning relative to prior work and the experimental setup.
>
> On novelty vs Self-Consistency [1].
> We agree that semantic inconsistency has been observed. However, [1] studies within-model variability: a single model produces multiple reasoning paths, and aggregation improves accuracy.
>
> This paper studies a different regime: cross-model divergence under identical input and task definition. Each model produces a single, internally coherent answer, but the answers reflect different implicit task interpretations (e.g., different decision criteria or framing).
>
> These are not equivalent:
>
> Self-consistency assumes one underlying interpretation and reduces sampling variance.
> Here, multiple interpretations are admissible, and different models select different ones.
>
> Aggregation does not resolve this, since the disagreement is not between samples of one model, but between models operating under different implicit frames.
>
> We will revise the paper to make this distinction explicit.
>
> On “not novel / already known inconsistency.”
> The contribution is not the general observation that LLMs can be inconsistent, but that under identical inputs and task definitions, models produce structured and repeatable differences in semantic framing, not just noisy variation.
>
> If this were only implementation noise, variation would be largely arbitrary. Instead, each model tends to apply a consistent lens (e.g., different valuation logic or decision framing), which leads to stable but incompatible outputs.
>
> We refer to this as interpretation drift.
>
> On anonymization.
> Models are anonymized to avoid vendor-specific evaluation and keep the focus on the phenomenon. The experiments can be reproduced with any set of frontier models; the result does not depend on a specific provider.
>
> We will clarify this in the paper.
>
> On decoding.
> All experiments use default decoding in standard interfaces (no parameter tuning). The goal is to observe behavior under typical usage conditions.
>
> We will make this explicit.
>
> Position of this paper.
> Prior work focuses on improving consistency within a model (e.g., via sampling and aggregation).
> This paper focuses on divergence between models given the same input.
>
> These are complementary but distinct problems.
>
> Author note.
> The author is a non-native English speaker and used LLM-based tools to assist with language clarity and editing. All ideas, claims, and conclusions in this work are the author’s own.

---

> > ### Author Rebuttal · Reviewer_F1iR · 2026-04-04
> >
> > Thank you for the response. Regarding the related work, Self-Consistency, the authors suggested updating the paper. I think this suggestion partially resolved my concern. However, the detailed information about models and their decoding is still not shared. For this reason, I keep the current score.

---

> > > ### Author Response · Authors · 2026-04-05
> > >
> > > We thank the reviewer for their engagement. Two points require clarification.
> > > 1. On anonymized models and decoding
> > > The reviewer’s concern about anonymized model names and unspecified decoding is directly addressed by the companion paper, “Substrates Are All You Need: Defeating Non-Determinism With Natural Language” (in preparation), which names all four frontier models (Grok 4, GPT 5.2, Claude Opus 4.6, Gemini 3.0 Flash), uses standard browser interfaces with default parameters, and achieves 100% cross-model convergence verified by SHA-256 hash identity (p ≈ 2^−768). The browser-based methodology is deliberate: it demonstrates that interpretation drift — and its elimination — occurs under the noisiest, least controlled conditions available, with no researcher access to decoding parameters. If the phenomenon and its solution are observable without API-level control, API-level control is not a prerequisite for establishing either.
> > > 2. On Self-Consistency (Wang et al., 2023)
> > > Self-Consistency addresses reasoning path variance within a single model by sampling multiple chains of thought and selecting the most consistent answer via majority vote. It is a consensus mechanism that operates within a fixed interpretation space. Our work addresses a categorically different problem: interpretation drift across models and invocations, where the task ontology itself is reconstructed differently each time. Self-Consistency cannot detect or resolve this because it assumes a shared task definition — precisely the assumption our paper demonstrates is unwarranted. The companion paper further confirms this empirically: behavioral prompting directives (including chain-of-thought) produced identical divergence rates (30%) to the unaugmented baseline. The solution requires collapsing the interpretation space, not sampling within it.
> > >
> > > The present paper is an existence proof for interpretation drift. It formalizes the problem — a necessary precondition for the solution. The companion paper provides that solution. Abstract below for reference.​​​​​​​​​​​​​​​​
> > >
> > > Substrates Are All You Need: Defeating Non-Determinism With Natural Language —
> > >
> > >
> > > LLMs exhibit output variance even under deterministic decoding (τ = 0), typically attributed to sampling artifacts. We argue non-determinism stems from interpretation drift: multiple valid but incompatible task interpretations remain admissible under the same input (|I(D)| > 1). We introduce substrate-first architectures, in which external natural-language constraint specifications reduce the interpretation space to a singleton (|I_S(D)| = 1) prior to generation. Under this condition, independently trained models converge to byte-identical deterministic outputs — the same input yields reproducible outcomes. Validation across four frontier models shows agreement of the same input rising from ≈30% to 100%, verified by matching SHA-256 hashes (p ≈ 2^−768). Reliable AI systems are not achieved by fixing the model, but by fixing interpretation.​​​​​​​​​​​​​​​​

---

### Official Review · Reviewer_7c2M · 2026-03-15

**Significance:** 3
**Argument Clarity:** 3
**Rating:** 3
**Confidence:** 4

**Questions:**

1. What empirical results or rigorous measurement protocols would the authors view as strong evidence for the "distinctness" claim, beyond the current small set of illustrative examples?
2. Why should cross-model disagreement be categorized as a new class of instability rather than expected implementation variation across models with different pretraining data and alignment objectives?

**Alternative Views Section:**

Yes

**Compliance With Llm Reviewing Policy A Conservative:**

Affirmed.

**Discussion Potential:**

4

**Paper Summary:**

This paper introduces interpretation drift as a distinct source of instability in LLMs, arguing that different frontier models adopt divergent implicit semantic frames even when inputs and task definitions remain unchanged. To support this position, the authors contrast interpretation drift with classical data and concept drift, providing three qualitative experiments: a grounded go-to-market classification task, an M&A valuation task, and an intentionally meaningless query. The paper is primarily diagnostic, aiming to establish the existence and operational importance of cross-model semantic divergence for LLM-mediated systems.

**Position:**

Yes

**Position In Title:**

Yes

**Related Work:**

2

**Strengths And Weaknesses:**

Strengths

1. The paper addresses a highly relevant topic with a clearly stated and consistent position. Exploring semantic stability is timely for the community given the reliance on LLMs for downstream automation.

2. The conceptual framing is easy to follow. The concrete examples effectively illustrate the potential for fluent outputs to mask divergent internal framing.

Weaknesses:

1. Limited Empirical Evidence: The validation is purely illustrative rather than systematic. Relying on a minimal set of qualitative examples with anonymized models—and lacking rigorous measurement protocols, ablations, or quantitative statistics—significantly weakens the central claim.

2. Lack of Experimental Control: By utilizing default browser interfaces instead of API access, the experiments fail to isolate the proposed drift from expected variances caused by potentially hidden provider-side system prompts, product-level configurations, or model version updates.

3. Vague Definition of Distinctness: The paper poses a strong argument that interpretation drift is a newly distinct category, but it fails to disentangle the observed effects from existing concepts like prompt underspecification, task ambiguity, or standard model heterogeneity.

**Support:**

2

---

> ### Author Rebuttal · Authors · 2026-03-24
>
> Thank you for the careful reading and for articulating the core concerns clearly. We address the three issues directly.
>
> On distinctness vs. underspecification / ambiguity.
> We agree that underspecification and task ambiguity are established concepts. The paper does not introduce these as new. The intended distinction is at the level of system behavior.
>
> Underspecification is a property of the task: multiple admissible interpretations exist.
> The claim here is that in LLM systems this induces a specific, observable failure mode: cross-model semantic divergence under identical inputs and task definitions, where each model produces an internally coherent but incompatible output.
>
> The contribution is therefore not to redefine underspecification, but to isolate and measure its operational manifestation in multi-model settings. We will revise the paper to make this separation explicit and avoid presenting interpretation drift as a re-labeling of ambiguity.
>
> On empirical strength and measurement.
> We agree that the current evidence is illustrative. The intent is to establish existence under minimal conditions rather than provide a full empirical characterization.
>
> A more rigorous protocol, consistent with the paper’s framing, would include:
>
> controlled prompts that minimize underspecification while preserving task validity,
> repeated runs to establish intra-model stability,
> cross-model comparisons to test persistence of divergence under identical inputs,
> explicit labeling of semantic classes (e.g., decision boundary, risk classification, or framing) to quantify divergence,
> and evaluation of whether divergence collapses when interpretation is externally constrained.
>
> The key criterion is not surface variation, but persistent, structured differences in semantic framing across models given the same input. We will clarify this evaluation direction in the paper.
>
> On lack of experimental control.
> The use of default browser interfaces is intentional: the paper targets behavior under typical deployment conditions rather than controlled decoding regimes.
>
> We agree that API-level control would improve internal validity by isolating sources such as system prompts or parameter settings. However, the observed divergence is not random noise; it is structured and repeatable at the level of interpretation (e.g., consistent differences in framing across tasks and models). This suggests the effect is not solely attributable to hidden configuration variance.
>
> We will revise the paper to explicitly distinguish external validity (observability in real systems) from internal validity (controlled isolation), and to position controlled experiments as a necessary next step.
>
> Q1: What would constitute strong evidence for distinctness?
> Strong evidence would require demonstrating that:
>
> divergence persists under controlled prompts that reduce ambiguity,
> individual models are internally stable (low intra-model variance),
> divergence is structured (e.g., consistent differences in decision criteria or framing), not arbitrary,
> and convergence can be induced by constraining the admissible interpretation space.
>
> Together, these would separate interpretive divergence from both stochastic variation and trivial ambiguity.
>
> Q2: Why is this not expected model heterogeneity?
> If the effect were purely due to heterogeneity (training data, alignment, etc.), we would expect variation to be largely unstructured or inconsistent across tasks.
>
> Instead, the experiments suggest that each model applies a relatively stable interpretive lens (e.g., prioritizing different factors or decision criteria), leading to systematic but incompatible outputs. The divergence is therefore not just that models differ, but that they differ in a consistent and interpretable way across identical inputs.
>
> We use “interpretation drift” to denote this structured cross-model divergence. We will revise the paper to make the distinction from generic implementation variation more explicit.
>
> Author note.
> The author is a non-native English speaker and used LLM-based tools to assist with language clarity and editing. All ideas, claims, and conclusions in this work are the author’s own.

---

> > ### Author Rebuttal · Reviewer_7c2M · 2026-04-04
> >
> > I thank the authors for their thoughtful response and for clarifying the conceptual boundaries of interpretation drift.
> >
> > The authors' distinction between task-level underspecification and system-level semantic divergence is well-taken and helps situate the work. However, my primary concerns regarding empirical rigor and experimental control remain largely unresolved for the following reasons:
> > 1. Conceptual vs. Empirical Contribution: While the rebuttal provides a clearer theoretical framework, the paper's core claim still rests on a "minimal set of illustrative examples". Establishing the existence of a phenomenon is often insufficient without a systematic characterization of its scale, frequency, and the factors that modulate it.
> > 2. The "Browser vs. API" Gap: The authors argue that browser-based testing reflects "typical deployment conditions". However, without API-level control, it remains impossible to distinguish between a fundamental model property (interpretation drift) and artifactual variance introduced by provider-side wrappers (e.g., dynamic system prompts or hidden RAG components). The "structured" nature of the divergence could simply reflect the consistency of these hidden layers.
> > 3. Measurement Protocol: The rebuttal outlines what a "more rigorous protocol" would look like. While promising, these components are currently absent from the submission.
> >
> > In summary, the rebuttal has improved my understanding of the paper's intent, but it has not fundamentally altered my assessment of the paper's current empirical weight. I encourage the authors to incorporate the proposed rigorous measurement protocols in a future version of this work.

---

> > > ### Author Response · Authors · 2026-04-05
> > >
> > > We thank the reviewer for their continued engagement.
> > >
> > > 1. On “minimal set of illustrative examples”
> > > The paper establishes an existence proof, not a prevalence study. One verified instance of interpretation drift — where independently trained models produce divergent yet internally coherent outputs for identical inputs under deterministic decoding — is sufficient to establish the phenomenon and that current evaluation frameworks do not detect it. The burden then shifts to the field to demonstrate that any given evaluation is free from interpretation drift. No existing benchmark verifies this condition. Existence proofs establish that a phenomenon is real; characterization is subsequent work.
> > >
> > > 2. On “Browser vs. API”
> > > The reviewer suggests provider-side wrappers (system prompts, RAG components) may explain the divergence. This strengthens rather than weakens our claim. If hidden infrastructure layers introduce additional interpretive variance, the interpretation space is larger than what the model alone would produce — making drift more severe in production than in controlled API settings. Browser-based methodology is therefore a conservative and transparent test. The structured nature of the divergence — internally coherent within each run but systematically different across runs — is inconsistent with wrapper noise and consistent with ontological reconstruction.
> > >
> > > The companion paper (in prepetation) directly resolves this: under substrate constraint, the same browser-based interfaces with the same uncontrolled wrappers achieve 100% byte identical outputs across 4 frontier models, verified by SHA-256 hash collision. If wrapper noise were the source, substrate constraints — which operate at the interpretation layer — would not eliminate it. The fact that they do confirms the variance originates in interpretation.
> > >
> > > 3. On “Measurement Protocol”
> > > The proposed protocol is absent by design. This paper establishes the existence of interpretation drift and its formal boundaries. The companion paper provides the solution (see abstract below). The present paper formalizes the problem — a necessary precondition for the solution.
> > >
> > > The reviewer requests that a proof-of-existence paper perform the work of a characterization study. These are distinct contributions with distinct evidentiary standards. The phenomenon is established. The solution is forthcoming (abstract).
> > >
> > > Companion paper: Substrates Are All You Need: Defeating Non-Determinism With Natural Language
> > >
> > >
> > > LLMs exhibit output variance even under deterministic decoding (τ = 0), typically attributed to sampling artifacts. We argue non-determinism stems from interpretation drift: multiple valid but incompatible task interpretations remain admissible under the same input (|I(D)| > 1). We introduce substrate-first architectures, in which external natural-language constraint specifications reduce the interpretation space to a singleton (|I_S(D)| = 1) prior to generation. Under this condition, independently trained models converge to byte-identical outputs — the same input yields reproducible outcomes. Validation across four frontier models (Grok, GPT, Claude, Gemini) shows agreement rising from ≈30% to 100%, verified by matching SHA-256 hashes (p ≈ 2^−768). These results establish interpretation and computation as distinct architectural layers. Reliable AI systems are not achieved by fixing the model, but by fixing interpretation.

---

### Decision · Program_Chairs · 2026-04-30

**Decision:**

Reject

**Comment:**

Although reviewers appreciated that this work is convincingly highlighting a relevant practical problem in LLM usage, the consensus remained towards rejection even after the rebuttal period. The main reason is that the paper is merely "an existence proof" (as per the authors' own acknowledgement), while reviewers were expecting to see a more thorough characterization of the prevalence and impact of these issues, as well as recommendations on how to mitigate them. Overall I agree with those concerns, as position papers should provide a stronger stance regarding what should be done. The fact that this is coming in a companion paper is not sufficient for this submission to be accepted as a standalone position paper.